



# Assessment of ACE-MAESTRO v3.13 multi-wavelength stratospheric aerosol extinction measurements

Sujan Khanal[1], Matthew Toohey[1], Adam Bourassa[1], C. Thomas McElroy[2], Christopher Sioris[3], Kaley A. Walker[4]

[1]Institute of Space and Atmospheric Studies, University of Saskatchewan, Saskatoon, Saskatchewan, Canada
[2] Department of Earth and Space Science and Engineering, York University, Toronto, ON, M3J 1P3, Canada
[3]Air Quality Research Division, Environment and Climate Change Canada, Toronto, Ontario, Canada
[4]Department of Physics, University of Toronto, Toronto, ON, M5S 1A7, Canada

*Correspondence to*: Sujan Khanal (sujan.khanal@usask.ca)

**Abstract.** The Measurement of Aerosol Extinction in the Stratosphere and Troposphere Retrieved by Occultation (MAESTRO) instrument on the SCISAT satellite provides aerosol extinction measurements in multiple solar wavelength bands. In this study, we evaluate the quality and utility of MAESTRO version 3.13 stratospheric aerosol extinction retrievals, from February 2004 – February 2021, through comparison with measurements from other satellite instruments. Despite

significant scatter in the MAESTRO data, we find that gridded median MAESTRO aerosol extinctions and stratospheric aerosol optical depth (SAOD) values are generally in good agreement with those from other instruments during volcanically quiescent periods. After volcanic eruptions and wildfire injections, gridded median MAESTRO extinction and SAOD are well-correlated with other measurement sets, but generally biased low by 40-80%. The Ångström exponent (AE), which can provide information on aerosol particle size, is derived from the MAESTRO spectral extinction measurements in the lowermost

stratosphere, showing perturbations after volcanic eruptions qualitatively similar to SAGE III for the eruptions of Ambae (2018) and Uluwan (2019). Differences in AE anomalies after the 2019 extratropical Raikoke eruption may be due to the different spatiotemporal sampling of the two instruments. Furthermore, we introduce a method to adjust MAESTRO extinction data based on comparison with extinction measurements from the Stratospheric Aerosol and Gas Experiment on the International Space Station (SAGE III/ISS) during the period from June 2017 to February 2021, resulting in improved

comparison during volcanically active periods. Our work suggests that empirical bias-correction may enhance the utility of MAESTRO aerosol extinction data, which can make it a useful complement to existing satellite records, especially when multi-wavelength solar occultation data from other instruments are unavailable.

**Short Summary.** Measurements of stratospheric aerosol from the MAESTRO instrument are compared to other measurements

to assess their scientific value. We find that medians of MAESTRO measurements binned by month and latitude show reasonable correlation with other data sets, with notable increases after volcanic eruptions, and that biases in the data can be





alleviated through a simple correction technique. Used with care, MAESTRO aerosol measurements provide information that can complement other data sets.

## 1 Introduction

Stratospheric aerosols play an important role in Earth's atmosphere and climate by modulating the Earth's radiation budget (Kremser et al., 2016); and references therein) and by influencing ozone depletion (Hofmann & Solomon, 1989; Solomon et al., 2022). Satellite measurements provide key information to characterize stratospheric aerosol properties and quantify their sources, which include volcanic eruptions (Bourassa et al., 2012; Vernier et al., 2011b), and wildfires (Bourassa et al., 2019; Khaykin et al., 2020, Hirsch and Koren, 2021). Satellite observations are essential in quantifying stratospheric aerosol

variability, its radiative forcing and impact on climate (Solomon et al., 2011; Friberg et al., 2018; McCormick et al., 1995; Stenchikov et al., 1998, Santer et al., 2014; Kloss et al. 2021).

Different techniques have been used to probe stratospheric aerosols from satellite observations. They include occultation (solar, stellar or lunar), limb scattering, limb emission, and lidar backscatter measurements. Satellite instruments that use the occultation method primarily use the sun as the source of light and measure the transmission of sunlight as the sun is observed

to rise and set from orbit (McCormick et al., 1979: Chu et al., 1989). They have provided an invaluable record of vertically resolved high-quality, stable, long-term aerosol optical properties, primarily extinction coefficient in narrow spectral bands. This is possible because occultation measurements are self-calibrating and have negligible bias due to long-term instrument deterioration (Lumpe et al., 1997). The use of a bright light source also makes it possible to achieve high signal-to-noise ratios in a relatively small instrument field of view, allowing measurements with a high vertical resolution. This has made solar

occultation measurements, particularly measurements from the Stratospheric Aerosol and Gas Experiment (SAGE) (McCormick, 1987; McCormick et al., 2020) series of instruments, the standard reference against which other measurements are compared for validation (Vernier et al., 2009: Rieger et al., 2019). Further, solar occultation measurements at different wavelengths provide information about aerosol particle size distribution, which is an important microphysical property that regulates radiative and chemical processes in the stratosphere (Lacis et al., 1992; Murphy et al., 2021).

Stratospheric aerosols have been observed from orbit since 1979 by different instruments using different techniques, and each with its own spatiotemporal sampling pattern. Merged data products combine different data sets, with the aim of producing a coherent description of the temporal and spatial evolution of aerosol physical and optical properties. Rieger et al. (2015) produced a merged aerosol data set based on SAGE II and OSIRIS aerosol extinction and applied a scaling to OSIRIS data in order to ensure consistency with the SAGE II record. The Global Space-based Stratospheric Aerosol Climatology (GloSSAC,

Kovilakam et al., (2020), Thomason et al., (2018)) provides climatologies of stratospheric aerosol properties spanning nearly 40 years. GloSSAC has been used in the construction of aerosol forcing fields for the Coupled Model Intercomparison Project (CMIP, Kovilakam et al. 2020, Rieger et al., 2020). Extinction coefficient measurements from the SAGE instruments are



central to the construction of GloSSAC, including SAGE II and SAGE III on the International Space Station (SAGE III/ISS, SAGE III hereafter). In the September 2005 - May 2017 gap between SAGE II and SAGE III measurements, the GloSSAC

climatology is constructed primarily based on single wavelength aerosol extinction measurements from the Optical Spectrograph and InfraRed Imaging System (OSIRIS) (Rieger et al., 2019) and the Cloud-Aerosol Lidar and Infrared Pathfinder Satellite Observation (CALIPSO) (Kar et al., 2019) instruments. More recently, the Climate data Record of Stratospheric aerosols (CREST, Sofieva et al. (2022)) reconstruction merges aerosol data from six satellite instruments: SAGE II, GOMOS and SCIAMACHY on Envisat, OSIRIS, OMPS on Suomi-NPP, and SAGE III /ISS.

The Measurement of Aerosol Extinction in the Stratosphere and Troposphere Retrieved by Occultation (MAESTRO) is a multi-wavelength solar occultation instrument that was launched into orbit in 2003 (McElroy et al., 2007) and remains operational at present. While some instruments (POAM III, Randall et al., 2001; SAGE III-Meteor, Thomason et al., 2007; GOMOS, Robert et al., 2016, Sofieva et al., 2024; SCIAMACHY, Malinina et al., 2018) have provided multi-spectral stratospheric aerosol measurements for portions of the period between SAGE II and SAGE III/ISS, MAESTRO is the only

such instrument in orbit that provides continuous data during the gap, overlapping with both instruments. Aside from some isolated cases related to volcanic eruptions (Sioris et al., 2010; Sioris et al., 2016), aerosol data from MAESTRO has not so far been widely used in scientific studies or multi-instrument merged data products. Nonetheless, MAESTRO data has the potential for important contribution to the long-term stratospheric aerosol record, especially during the gap between SAGE II and SAGE III, if the measurements are of sufficient quality. Since it overlaps with both SAGE II and SAGE III observations,

comparisons between them can reveal key features in MAESTRO data.

In this study, the aim is to evaluate the quality and utility of MAESTRO measurements of stratospheric aerosol extinction through comparison with measurements from other satellite instruments. We also explore methods to reduce observed biases and scatter in MAESTRO aerosol extinction data, aiming to enhance its utility for scientific analysis and potential data merging.

**2 Data and Methods**

**2.1 Data**

**2.1.1 MAESTRO**

MAESTRO (Bernath et al., 2005; McElroy et al., 2007) is a dual optical spectrophotometer that is part of the Atmospheric Chemistry Experiment (ACE) on the SCISAT satellite. It is a Canadian-led mission mainly supported by the Canadian Space

Agency. It was launched into a low Earth circular orbit in August 2003 at an altitude of 650 km and an inclination of 74°. MAESTRO makes measurements primarily in the solar occultation mode, at different tangent heights within the latitude range 85° S to 85° N. MAESTRO makes up to 15 sunrise and 15 sunset measurements each day and has a vertical resolution of 1-2



km. SCISAT also carries another instrument which is a high spectral resolution Fourier Transform Spectrometer (FTS) operating in the infrared region from 2.2 to 13.3 μm. ACE-FTS measurements provide vertical profiles of temperature and

many trace gases with a vertical resolution of 3-4 km (Bernath, 2017). MAESTRO and ACE-FTS share a suntracking mirror and thus make collocated observations.

The nominal MAESTRO wavelength range is 515-1015 nm for the visible spectrometer. There are absorption features in the MAESTRO spectral measurements due to ozone, nitrogen dioxide, water vapour and oxygen, and contribution due to scattering by molecules and aerosols (McElroy et al., 2007). Profiles of ozone, nitrogen dioxide and optical depth are retrieved from the

MAESTRO transmission spectra as a function of altitude, using a modified differential optical absorption technique followed by an interactive Chahine relaxation inversion algorithm (McElroy et al., 2007, and references therein).  The pressure and temperature data used for the retrieval are obtained from the ACE-FTS measurement from the same occultation, as the two instruments measure simultaneously.  The MAESTRO version 3.13 processor uses ACE-FTS version 3.5/3.6 pressure and temperature profiles (Boone et al., 2013) which ends in February 2021.

Aerosol extinction can be retrieved after accounting for molecular absorption and scattering. In this study, we use the MAESTRO version 3.13 aerosol extinction coefficients (525, 530, 560, 603, 675, 779, 875, 922, 995 and 1012 nm), which are reported every 0.32 km, from February 2004 to February 2021. MAESTRO version 3.12 aerosol extinction was compared to AERGOM retrievals from the GOMOS instrument, suggesting that MAESTRO had a high bias through the stratosphere (Robert et al., 2016). We use the temperature profile information from the ACE-FTS to get the lapse rate tropopause height

based on the World Meteorological Organization criteria (WMO, 1992). This allows the stratospheric component of the MAESTRO aerosol extinction coefficient profile to be separated for further analysis. Cirrus cloud screening is not performed as part of the MASTRO data product.

### 2.1.2 SAGE II

The SAGE II (McCormick, 1987) instrument was launched in October 1984 on the Earth Radiation Budget Satellite (ERBS)

and was operational until 2005. ERBS orbited the Earth at an altitude of 610 km and had an inclination of 57°, which caused its orbital plane to precess with respect to the sun. SAGE II was a solar occultation instrument with seven channels centered at 385, 448, 453, 525, 600, 935, 1020 nm. About 32 occultations were made per day until mid-2000, after which only 16 measurements were made per day. Depending on the season, it made measurements between approximately 80° N and 80° S. In this study, we use version 7 of the SAGE II data product (Damadeo et al., 2013), which includes cloud-screened aerosol

extinctions at 385, 453, 525, and 1020 nm with vertical resolution of 1 km that are reported at every 0.5 km height interval.





### 2.1.3 OSIRIS

OSIRIS (Llewellyn et al., 2004) is a limb scatter instrument launched in 2001 on board the Odin satellite. Odin was placed in a sun-synchronous orbit at an altitude of 600 km and an inclination of 98°. This orbit allows OSIRIS to sample latitude ranging from 82° S to 82° N around the equinoxes while sampling is restricted to the summer hemisphere around the solstices. The
OSIRIS spectrograph measures wavelengths between 284 and 810 nm with approximately 1.0 nm resolution, scanning at different tangent altitudes. These measurements provide vertical sampling every 2 km with a vertical resolution of approximately 1 km. Compared to occultation measurements, limb scattering provides a greater sampling frequency, which can reach up to 400 observations per day, depending on the time of the year and location. In this study, the latest version 7.2 of the OSIRIS aerosol is used, which provides cloud-screened vertical profiles of aerosol extinction coefficients at 750 nm
(Rieger et al., 2019).

### 2.1.4 SAGE III

SAGE III on ISS began its mission in June 2017 (McCormick et al., 2020). The ISS orbit's inclination is 51.6° and maintains an average altitude of around 400 km. The SAGE III makes observations of stratospheric aerosol extinction coefficient at wavelengths ranging from 385 to 1550 nm with latitude coverage between roughly 70° S and 70° N. Similar to SAGE II, it
uses the solar occultation technique to retrieve vertical profiles of multi-wavelength aerosol extinction coefficient (384, 449, 521, 602, 676, 756, 869, 1022, and 1544 nm). Here, we use version 5.2 of the SAGE III aerosol extinction vertical profile data (Kovilakam et al., 2023), which has a vertical resolution of about 1 km and is reported every 0.5 km. No cloud screening is included in the released SAGE III data.

### 2.2 Sampling Coverage

Figure 1 depicts the frequency of observations as a function of latitude and time for MAESTRO, SAGE II, SAGE III and OSIRIS measurements over a two-year period. MAESTRO and SAGE III observations are from 2018 and 2019 whereas SAGE II and OSIRIS observations are from 2002 and 2003. The observations are binned monthly in 10° latitude intervals. It shows that MAESTRO samples the high latitudes well, with more than 100 occultation events in some bins poleward of 50°. However, its sampling over the tropics is quite sparse. On the other hand, SAGE III has denser coverage in the tropics whereas high
latitudes are not sampled regularly. This indicates that MAESTRO is particularly well-suited to study high latitude volcanic eruptions and wildfires, thus providing complementary information to SAGE II and SAGE III. OSIRIS, which makes limb scattering measurements, offers higher number of observations by nearly an order of magnitude, but there are significant gaps over the extra-tropics around winter months.



**Figure 1: Number of measurements per month and 10° latitude bin by the MAESTRO (top left), SAGE II (top right), SAGE III (bottom left) and OSIRIS (bottom right) instruments for a two-year period. Note the different scale for the OSIRIS observations.**

## 2.3 Methods

Data from each of the four instruments include profiles of geolocated aerosol extinction coefficient. For each profile, only measurements above the tropopause are considered. Apart from MAESTRO profiles, for which the tropopause information is determined from the collocated ACE-FTS measurements, tropopause information for other instruments is provided as part of the scientific data product. For SAGE II analysis, measurements from two wavelengths at 525 and 1020 nm are used. For MAESTRO and SAGE III analysis, six pairs of approximately matched wavelengths were selected, with wavelengths of 525, 603, 675, 779, 875 and 1012 nm for MAESTRO and 521, 602, 676, 756, 869 and 1022 nm for SAGE III. During background stratospheric conditions (relatively undisturbed by volcanic eruptions or wildfires), the difference in extinction is expected to reach around 6% for the pair having the largest separation in wavelengths at 779 nm and 756 nm and less than 3% for all other



wavelength pairs. Since this is a relatively small change, we ignore any difference in extinction coefficient values due to the difference in corresponding wavelengths between the two instruments.

First, MAESTRO extinction is linearly interpolated to 0.5 km height intervals to match the vertical grid spacing of SAGE II
and SAGE III data. Then, stratospheric aerosol optical depth (SAOD) at each wavelength is calculated by vertically integrating the respective extinction profile from the tropopause upward to the top of the measured profile. Multi-wavelength measurements from the occultation instruments also allow for the calculation of Ångström Exponent (AE) at each altitude level. AE is a measure of the wavelength dependence of extinction and is related to the aerosol particle size distribution (Ångström, 1964; Eck et al., 1999; Malinina et al., 2019). It can be calculated by determining the slope of a linear fit between
the logarithm of extinction coefficients ($\beta$) and the logarithm of wavelengths ($\lambda$) (Mironova et al., 2012), as shown in equation 1, where AE is denoted by $\alpha$

$$\alpha = -\frac{d\ln\beta}{d\ln\lambda},\qquad(1)$$

Extinction measurements at five corresponding wavelengths (525 nm is excluded, details in Sect. 3.1) are used to calculate AE for MAESTRO and SAGE III respectively at each altitude of each profile, using the ordinary least squares solution to get the
slope from the linear regression in log space. For SAGE II, extinction coefficient measurements at only the two wavelengths are used.

Aerosol extinction coefficient, SAOD and AE are then binned in regular temporal (monthly) and spatial (10° latitude) grids for each dataset. To minimize the impact of outliers, we use the median of the measurements within each bin as an estimate of the distribution centre. Since some of the outlier values could be due to the presence of clouds in MAESTRO and SAGE III
datasets (which are not screened for cloud contamination), this step also ensures that the impact of outliers arising from stratospheric clouds is minimized. Thus, a gridded dataset for each instrument as a function of time, altitude, latitude and wavelength is produced.

## 3 Results

### 3.1 Extinction

Figure 2 shows the median and standard deviation of extinction coefficients from the SAGE II, OSIRIS, MAESTRO and SAGE III gridded data at an altitude of 15.5 km in the midlatitudes of the Southern Hemisphere (SH) and Northern Hemisphere (NH). Although SAGE II measurements are available from 1985, only data from 1998 is shown when the impact of the 1991 Pinatubo eruption had mostly subsided. SAGE II extinction at 750 nm is derived by linearly interpolating 525 and 1020 nm extinction coefficients in a log extinction-log wavelength space. MAESTRO and SAGE III extinctions coefficients values are
shown at native 779 and 756 nm respectively. Volcanic eruptions and wildfires are noticeable in the extinction coefficient





timeseries in both hemispheres. There is a good degree of similarity between SAGE II, OSIRIS and SAGE III measurements during data overlap periods. Even though the MAESTRO extinction coefficient time series exhibits more scatter, it shows variations qualitatively consistent with other measurements, including clear increases in extinction following major volcanic eruptions and wildfires. During quiescent periods, the magnitude of MAESTRO extinction coefficients matches well with that

of the SAGE instruments and OSIRIS. However, MAESTRO underestimates peak extinction values after major volcanic eruptions and wildfires by a factor of 2 or more. For example, extinction from SAGE and OSIRIS is larger than MAESTRO following the 2019 Raikoke and 2019-2020 Australian wildfires even after accounting for the data variability as indicated by the standard deviation in Fig. 2. Despite that, the MAESTRO aerosol extinction coefficient measurements are correlated with that from the other instruments.


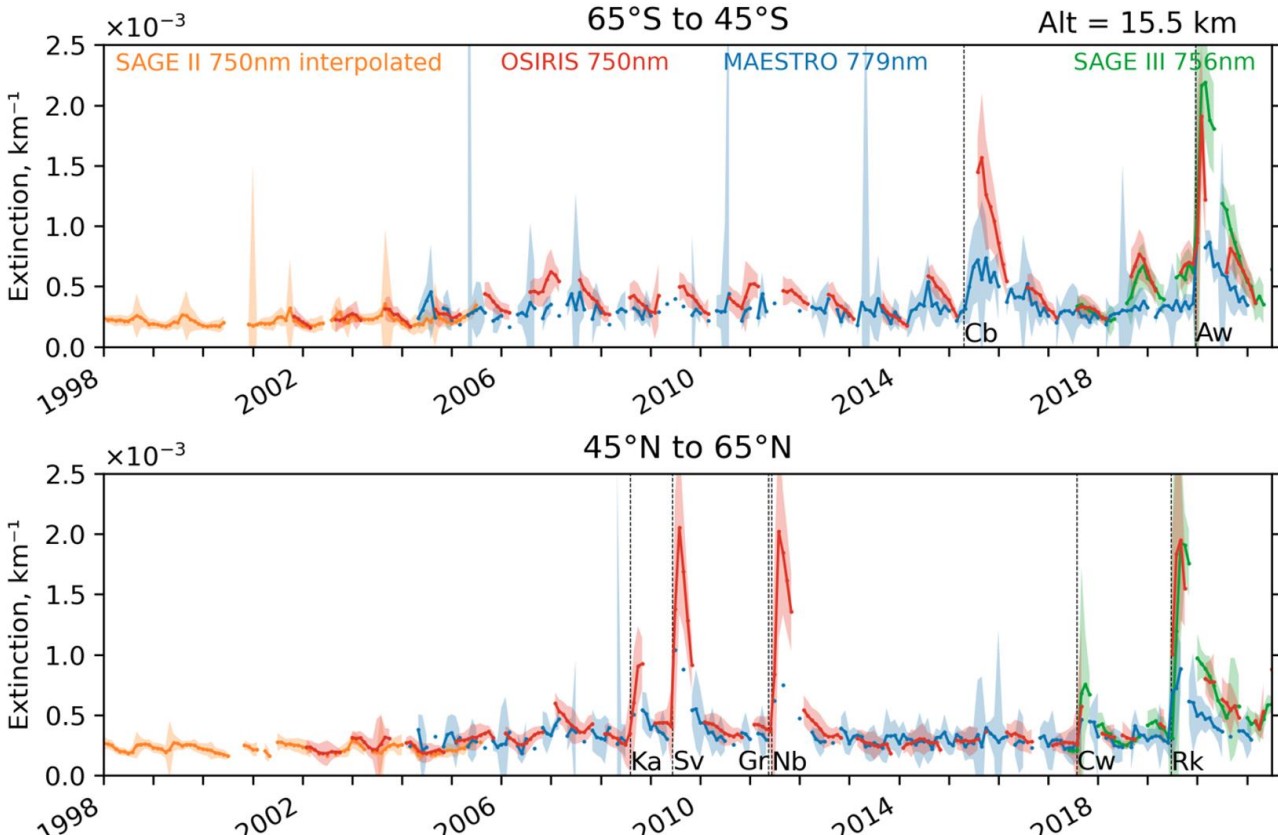

**Figure 2: Comparison of monthly zonal median extinction coefficients measured by SAGE II, OSIRIS, MAESTRO, and SAGE III at an altitude of 15.5 km in the midlatitudes of the (top) SH and (bottom) NH. Shaded area represents ± one standard deviation.**
**Data in two 10-degree latitude bins are combined to show the timeseries by calculating the average of the medians. Dotted vertical lines correspond to the time when notable volcanic eruptions or wildfires occurred within or near respective latitude ranges. They represent Calbuco (Cb, 41°S) and the Australian Wildfires (Aw) in the SH and Kasatochi (Ka, 52°N), Sarychev (Sv, 48°N), Grimsvotn (Gr, 64°N), Nabro (Nb, 13°N), the Canadian Wildfires (Cw) and Raikoke (Rk, 48°N) in the NH. Although Nabro was a**





**tropical eruption, it is shown here because it impacted the high latitudes in the NH. Note the multiplicative factor of 10$^{-3}$ (shown**
**above y-axis) used to get the extinction coefficients in units of 1/km.**

Figure 3 depicts the median of all extinction measurements made by MAESTRO and SAGE III averaged over their overlap period from June 2017 to February 2021, plotted as a function of latitude and height. The start of this overlap period represents relatively clean background conditions with the Canadian wildfires being the only major event that impacted stratospheric aerosol levels in the NH. But after mid-2018, a number of events such as Ambae, Ulawun, Raikoke and the Australian wildfires

of 2019/2020 caused significant perturbations in the stratospheric aerosol levels. Figure 3 compares extinctions for four out of six common wavelength pairs between the two instruments. Results from both instruments show similar qualitative features of the time-averaged stratospheric aerosol distribution, with maximum values in the lower stratosphere of each hemisphere and decreasing extinction above ~20 km. MAESTRO extinction at 525 nm shows a pronounced peak between altitudes of 10 to 13 km in the extratropics of both hemispheres. For other wavelengths, the peak values occur at lower altitudes and decrease

gradually with height. For SAGE III however, the peak extinction values in tropics and mid-latitudes occur a few kilometers above the tropopause before they start to decrease with height. The percentage difference plot (Fig. 3, third column) highlights the high bias in MAESTRO extinction at 525 nm around 12 km. Currently, it is not known what causes this unique feature at this wavelength, but it is absent at the other five wavelengths. The figure also shows that MAESTRO extinction at shorter wavelengths has a low bias of 40-80% compared to SAGE III nearly everywhere in the lower stratosphere except right above

the tropical tropopause region. The correlation between the two measurements (Fig. 3, fourth column) is mostly greater than 0.6 in this broad region.



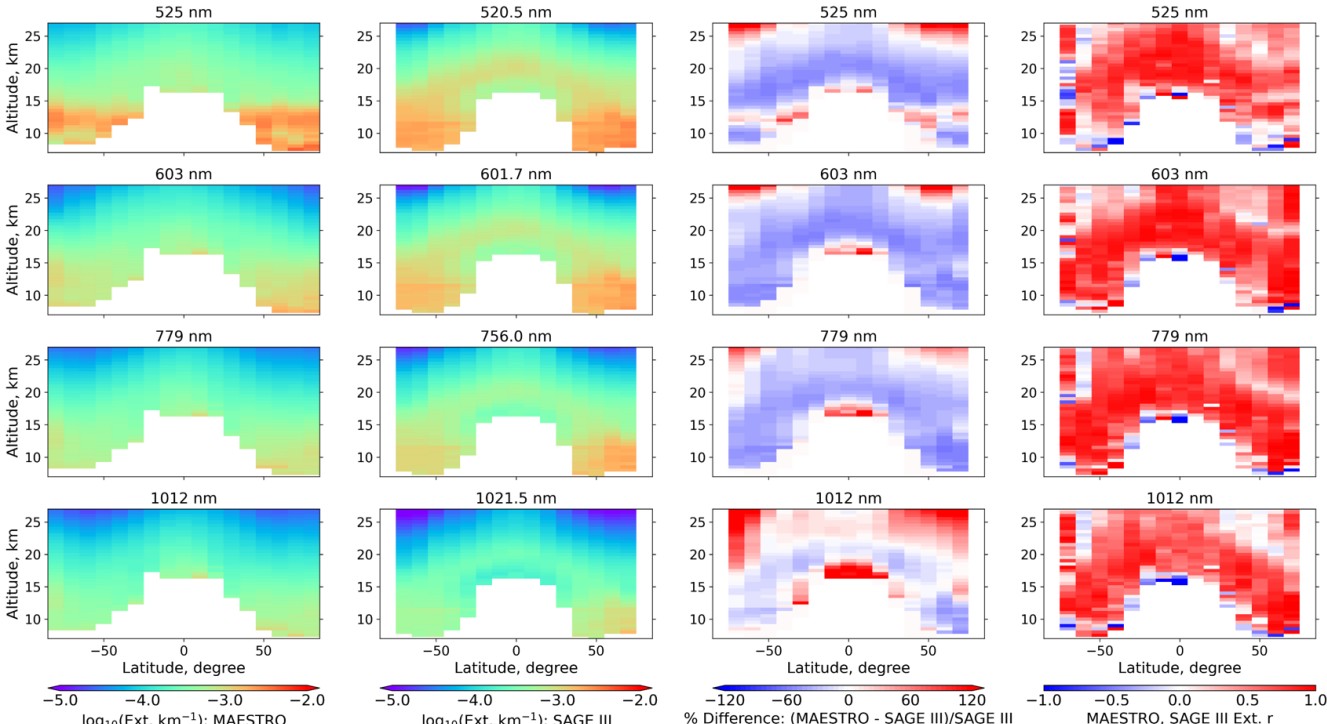

**Figure 3: Median extinction coefficient from the full MAESTRO (first column) and SAGE III (second column) data sets at four different wavelengths during their overlap period from June 2017 to February 2021, shown as a function of altitude and latitude. The data is binned in 10° latitude bins. The third column shows the percentage difference (MAESTRO − SAGE III)/SAGE III in extinction coefficients compiled from the two instrument data sets. The fourth column shows the Pearson's correlation coefficients between MAESTRO and SAGE III extinction coefficients.**

## 3.2 SAOD

Figure 4 shows monthly median SAOD derived from the MAESTRO and SAGE III extinctions as a function of latitude and time at three wavelengths. Due to its orbital characteristics, as was also seen in Fig. 1, MAESTRO has large data gaps in the tropics and is better suited for investigating extra-tropical volcanic eruptions and wildfires. Figure 4 shows the zonal distribution of SAOD during MAESTRO's overlap period with SAGE III, that included Canadian wildfires, Raikoke and the Australian wildfires as the three largest extra-tropical events. Signal from these three events and the two tropical eruptions of Ambae and Ulawun are evident in SAGE III SAOD data. The three extra-tropical events are also evident in MAESTRO SAOD, which shows strong and persistent enhancements for Raikoke and the Australian wildfires poleward of 50°, a region which is not well-sampled by SAGE III. It is also evident that the peak in SAOD values from MAESTRO for all wavelengths is lower than those from SAGE III at corresponding wavelengths. However, there is a general agreement in SAOD magnitude between the two sets of measurements during quiescent periods, despite the larger scatter in MAESTRO data. The Pearson's correlation coefficient between 779 nm MAESTRO and 756 SAGE III SAOD over the set of months and latitude bins where both



instruments have measurements is 0.83 and the root mean square difference is 0.00367, which corresponds to a relative underestimation of 32% by MAESTRO.

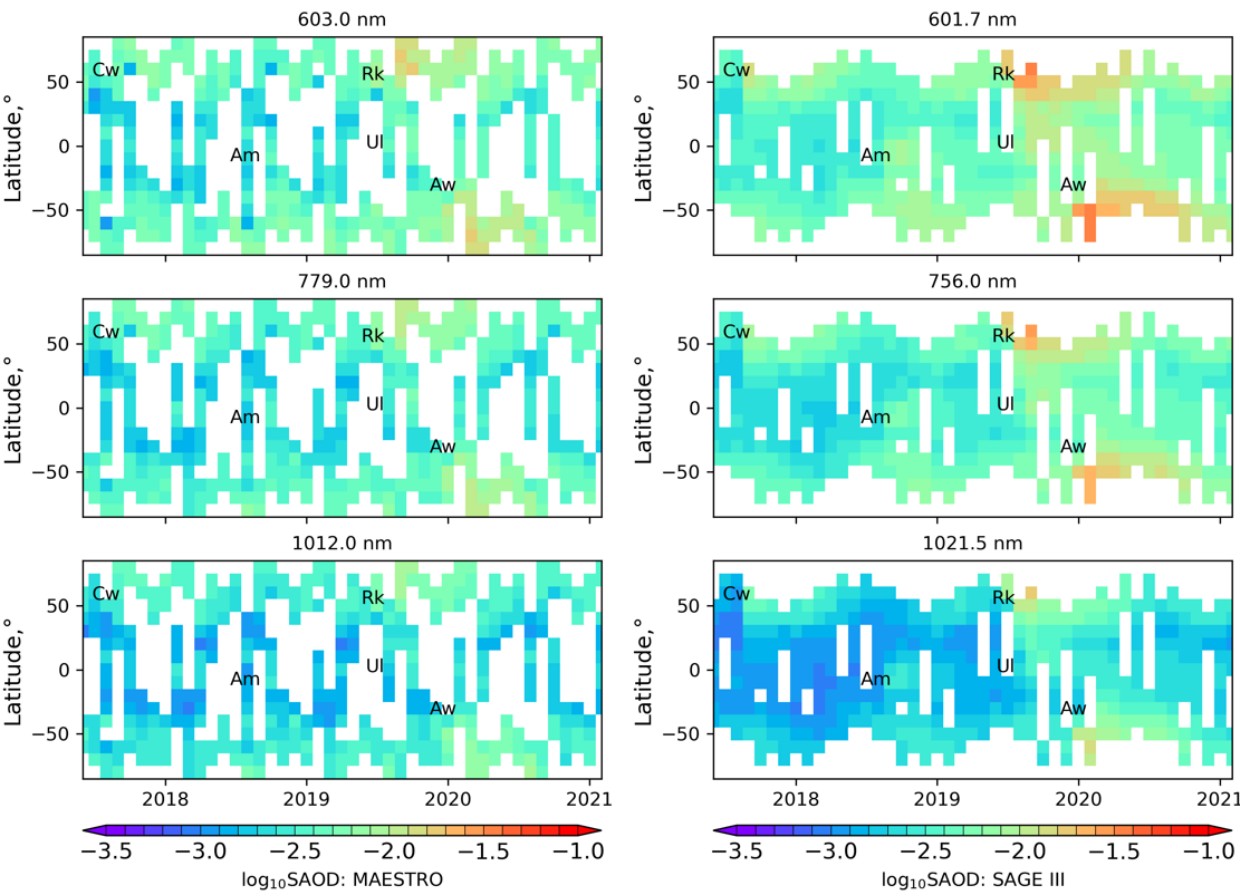

**Figure 4: Monthly median SAOD derived from MAESTRO (left panel) and SAGE III (right panel) extinctions at three different wavelengths as a function of latitude and time. Labels Cw, Am, Rk, Ul and Aw mark the latitude and dates of the Canadian wildfires, the Ambae, Raikoke and Ulawun eruptions, and the Australian wildfires, respectively.**

### 3.3 Ångström Exponent

One of the major advantages of satellite-based solar occultation instruments is that they can provide measurement of aerosol extinction at multiple wavelengths. The AE, which reflects this spectral relationship, contains valuable information about aerosol particle sizes (Malinina et al., 2019; Schuster et al., 2006). As aerosol content of the stratosphere varies with time or location, we expect the aerosol extinction to vary simultaneously for all wavelengths, by different amounts corresponding to the size distribution of the aerosols. As a result, measurements with a high signal to noise ratio should show strong correlation between wavelengths: for example, the correlation of SAGE III extinction measurements is greater than 0.9 for all wavelength pairs in the stratosphere (Fig. S1). Figure 5 illustrates the spectral correlation of the MAESTRO gridded median extinctions





over the full time range. It shows the Spearman's correlation coefficient (ρ) between 603 nm and four other wavelengths as a function of altitude and latitude. The correlation is above 0.7 for all latitudes and heights between extinctions at 603 and 675 nm. The correlation usually decreases as the wavelength separation gets larger. All four wavelength pairs shown in Fig. 5 have higher correlation in a region 3-10 km from the tropopause, and lower correlation above this region. This suggests that the confidence in calculated AE values will be higher in this region of the lower stratosphere. Furthermore, even though spectral correlation between wavelengths that are close to each other is reasonably high, they span a relatively small range in the wavelength space such that even minor uncertainty in the measurement of extinction at any wavelength can lead to large uncertainty in AE values.

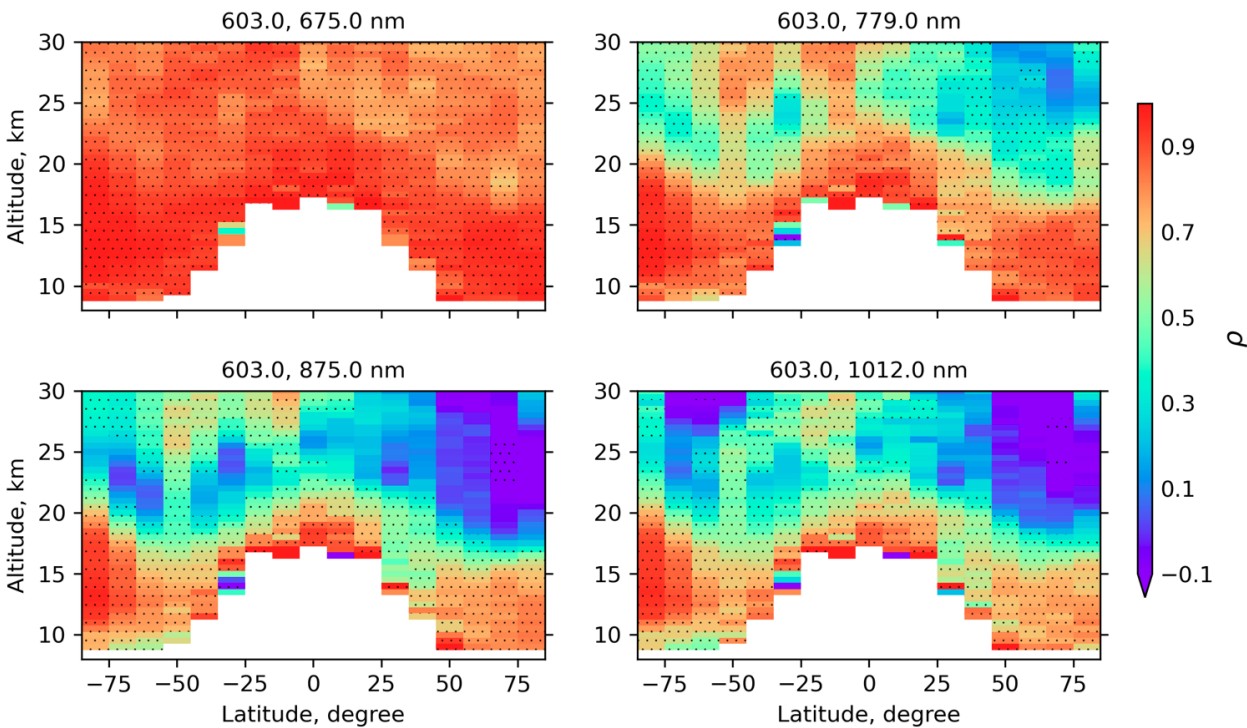

**Figure 5: Spearman's correlation coefficient of MAESTRO extinction coefficients between 603 nm and four other wavelengths. The four panels represent different wavelength pairs as indicated by the values on the top of each panel. Dots represent regions where the correlation is significant at 99% confidence level.**

Time series of AE in the lower stratosphere based on extinction measurements from MAESTRO and SAGE III are shown in Fig. 6. We show the AE at 12 km in the lowermost stratosphere, where correlations are typically strongest between MAESTRO wavelengths (Fig. 5) and where aerosol perturbations from the moderate eruptions and wildfires are most pronounced. On average, MAESTRO AE has a low bias of magnitude around 1, compared to SAGE III. Despite large variability, MAESTRO shows positive perturbations in AE values in the SH mid-to-high latitudes following the tropical Ambae and Ulawan eruptions, in agreement with the SAGE III results. In the NH, MAESTRO AE also shows increases in the high latitudes after the Canadian wildfires and the Raikoke eruption—in both cases in apparent contrast to the SAGE III results which show apparent decreases



of AE immediately after these aerosol events. The SAGE III AE decrease after Raikoke has been interpreted as signaling an increase of the particle size after this eruption, setting it in contrast to other recent eruptions which produce positive AE anomalies suggesting particle size decreases (Wrana et al., 2023, Thomason et al., 2021). We note that in the first ~6 months after the Raikoke eruption, SAGE III measurements are almost entirely equatorward of 50N, while MAESTRO observations are mostly limited to poleward of 50N, which may explain the apparent inconsistency in AE results from the two instruments.

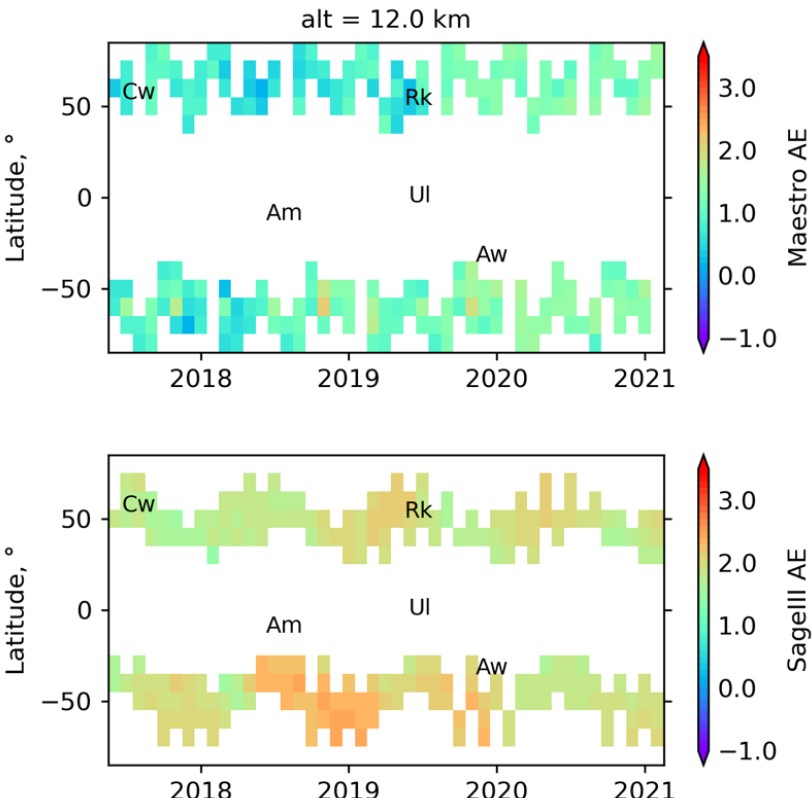

**Figure 6: Monthly median AE at an altitude of 12 km derived from MAESTRO (top panel) and SAGE III (right panel) extinctions plotted as a function of latitude and time. Labels Cw, Am, Rk, Ul and Aw represent Canadian wildfires, Ambae, Raikoke, Ulawun and Australian wildfires respectively.**

## 4 Post-processing MAESTRO extinction measurements

While the MAESTRO aerosol extinction data contains significant variability, results from the previous section suggest that with sufficient sampling and use of robust statistics like median values, the data are reasonably correlated with the highly reliable measurements from SAGE III, which suggests that MAESTRO data contain useful information. In this section, we explore two potential methods that can lower the observed biases and noise in MAESTRO extinction measurements. Details about these two approaches are provided in the following discussion.





## 4.1 MAESTRO extinction tuning

To account for the wavelength-dependent bias in MAESTRO extinction measurements (Sect. 3.1), a "tuning" approach based on comparisons with the SAGE III measurements is implemented. Empirical correction factors are constructed to remove observed biases from MAESTRO measurements based on the observed relationship between MAESTRO and SAGE III binned

median extinction data. Similar scaling procedures have been used to improve agreement between OSIRIS and SAGE extinction values (Rieger et al., 2015). For each wavelength and at every altitude bin, the MAESTRO and SAGE III data are related using a power-law function of the form $y = ax^b$, using non-linear least squares approach, where $a$ and $b$ represent the scaling and exponent parameters, and $x$ and $y$ represent SAGE III and MAESTRO aerosol extinctions respectively. Since extinction measurements can span orders of magnitude, using a power law fit (or equivalently, a linear fit in log-space) helps

ensure a fit that works for the full range of data—a linear fit tends to be heavily weighted by the largest extinction values. Tuned MAESTRO extinction coefficients are then computed by inverting the power-law relation. This correction method is applied for the entire MAESTRO extinction measurements and SAOD and AE values are re-calculated using adjusted values.

SAGE III instead of SAGE II was picked as the benchmark because the overlap between SAGE III and MAESTRO covers nearly four years that include both volcanically quiescent and active periods, and therefore the extinction values span a

relatively large range. Gridded extinction coefficients from the entire overlap period between MAESTRO and SAGE III are compared using a scatterplot in log-scale at every altitude bin and at each of the six common wavelength pairs. To ensure robust measurement signal, only bins with at least 10 valid extinction retrievals for each instrument were included in the comparisons. The two panels in Fig. 7 show example scatterplots of extinction coefficients at two different wavelengths and at two different altitude levels. Comparison seen in the left panel in Fig. 7 has high correlation (0.87), and majority of data

points lie close to the regression line. This indicates that the two parameters from the power-law fit can correct the bias in MAESTRO extinctions reasonably well. On the other hand, the right panel represents an example (correlation 0.45) that is challenging for the correction approach. For this particular latitude and altitude, the overall scatter is larger, most likely a result of larger random error in the MAESTRO extinction data at this wavelength and altitude. There is also a subset of data with small SAGE III values and relatively large MAESTRO values, which notably affects the best fit line away from the slope of

the majority population of points. Similar analysis was performed for each of the six common wavelengths and at every altitude bin. This results in two power-law fit parameters and the correlation coefficient as a function of altitude for each wavelength, which is shown in Fig. 8. Correlations are greater than 0.7, except for an altitude range of 16-19 km at longer wavelengths, where they are on the order of 0.5 to 0.6. The parameters are relatively uniform with altitude and have close similarity for adjacent wavelengths.





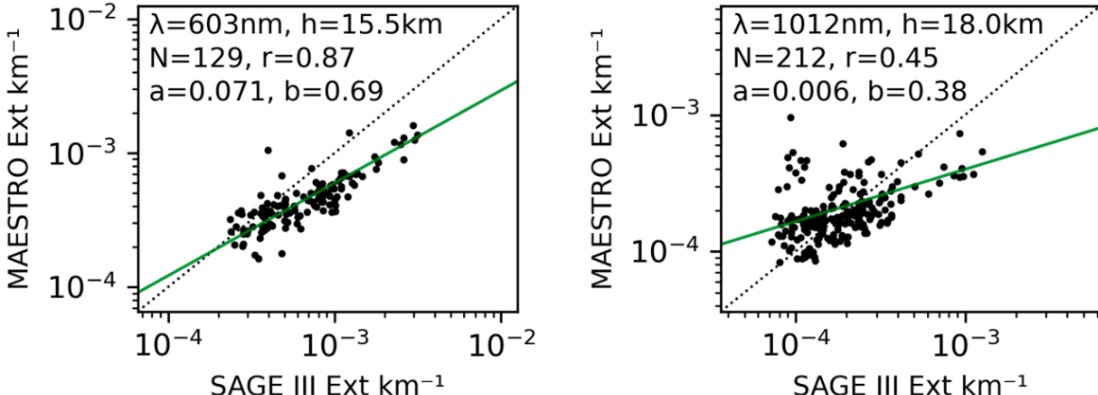

**Figure 7: Scatterplot in log-scale showing the MAESTRO and SAGE III median binned extinction coefficients at 603 nm and an altitude of 15.5 km (left panel) and at 1012 nm and an altitude of 18.0 km (right panel). Number of data points and the Pearson's correlation coefficient is also shown. Green line represents the power-law regression line whose scaling and exponent parameters are given by 'a' and 'b' respectively. The dotted black line is the one-to-one line.**


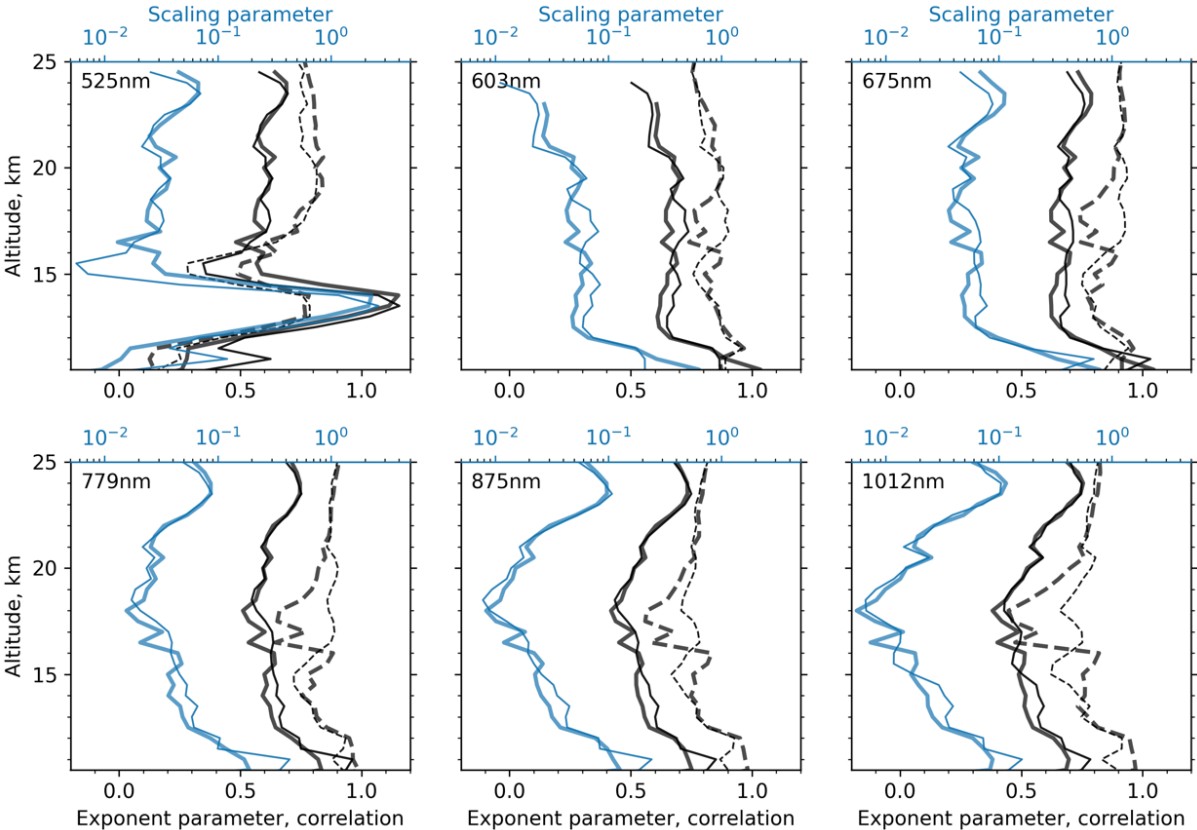

**Figure 8: Comparison metrics as a function of altitude for the six common wavelength pairs between MAESTRO and SAGE III. Correlation coefficient (thick dashed black line) and exponent parameter (thick solid black line) from the power-law fit are plotted using the bottom horizontal axis, and the scaling parameter (thick solid blue line) from the power-law fit is plotted along the top horizontal axis in log-scale. Thin lines represent corresponding metrics but with trimmed MAESTRO data in relation to Rayleigh scattering correction (see Sect. 4.2).**




The impact of correcting MAESTRO extinction values is shown in Fig. 9 with an example from the lower stratosphere in the northern midlatitude region. The top panel in Fig. 9 reveals that the correction makes the peak in MAESTRO extinctions align better with SAGE III following major volcanic eruptions and wildfires during the period of overlap, and furthermore with

OSIRIS within the SAGE gap period. The comparison during quiescent period remains roughly similar. Clear signals from the Kasatochi, Sarychev, Grimsvötn and Nabro eruptions, that occurred after SAGE II and before SAGE III operations, are seen in the monthly time-series data. Even though there are only a few data points, the peak in adjusted extinction values matches those from the OSIRIS quite well. Mid-to-high latitude SAOD values derived from adjusted MAESTRO extinctions in Fig. 9 (middle panel) shows improved comparison with OSIRIS and SAGE III following volcanic eruptions and wildfires.

The bottom panel in Fig. 9 shows the comparison of AE values in the NH mid latitudes and 12.0 km altitude calculated from measurements by different instruments including AE calculated from the tuned MAETRO extinctions. AE values using corrected extinctions from MAESTRO show slightly reduced discrepancies with SAGE III compared to the same from the uncorrected data. However, the AE from MAESTRO exhibits large scatter, and it is difficult to clearly identify disturbances from the background values. Figure 9 shows the increase in AE suggested by MAESTRO after the Raikoke eruption is in

contrast to the SAGE III results as discussed above, which may be due to the different latitudinal sampling of the two instruments. Moreover, a gradual decreasing trend in AE values between its start in 2005 and about 2019 is also noticeable in the MAESTRO timeseries. Further investigation shown this trend is mostly due to the decreasing trend in MAESTRO extinctions at shorter wavelengths, mainly 603 and 675 nm (Fig. S2). AE values from the end of the SAGE II record and the beginning of the SAGE III record show a small difference, suggesting the MAESTRO AE trend may be an artefact. On the

other hand, noting the large scatter in the MAESTRO AE values and the different sampling pattern compared to the SAGE instruments, further analysis would be needed to determine if the apparent differences between MAESTRO and SAGE AE have a geophysical or instrumental origin. Latitude-time plots of the tuned MAESTRO SAOD and AE are shown in Fig. S3 and Fig. S4 respectively.



**Figure 9: Comparison of median stratospheric aerosol extinction coefficients (top panel), SAOD (middle panel) and AE (bottom panel) between measurements from different instruments in northern midlatitudes. Shaded area represents ± one standard deviation. Data in two 10-degree latitude bins are combined to show the time series by calculating the average of the medians. There is no AE data from OSIRIS because it is based on single 750 nm measurements. Selected wavelengths for each instrument are labelled. Selected altitude for extinction and AE plots is also labeled on the top right of each plot. For MAESTRO, values before and after correction are shown.**



## 4.2 Impact of Rayleigh Scattering Correction

A potential reason for the scatter in MAESTRO data is due to the treatment of Rayleigh scattering. Tangent altitudes for each
measurement are a retrieved quantity incorporating measurements made by the ACE-FTS instrument, which shares the same
line of sight as MAESTRO, and the altitude of the lowest retrieved tangent altitude varies from profile to profile. The frequency
distribution of that FTS lowest tangent altitude (or cutoff altitude) is shown in Fig. 10. If the cutoff altitude is above 10 or 15
km, then the calculated air column that is used to remove Rayleigh scattering may be inaccurate, negatively affecting the
accuracy of MAESTRO retrievals. However, if the cutoff altitude is lower, then this is not an issue. SCISAT loses its lock on
the Sun for tangent heights below ~5 km, which is the lower limit of the ACE-FTS data.

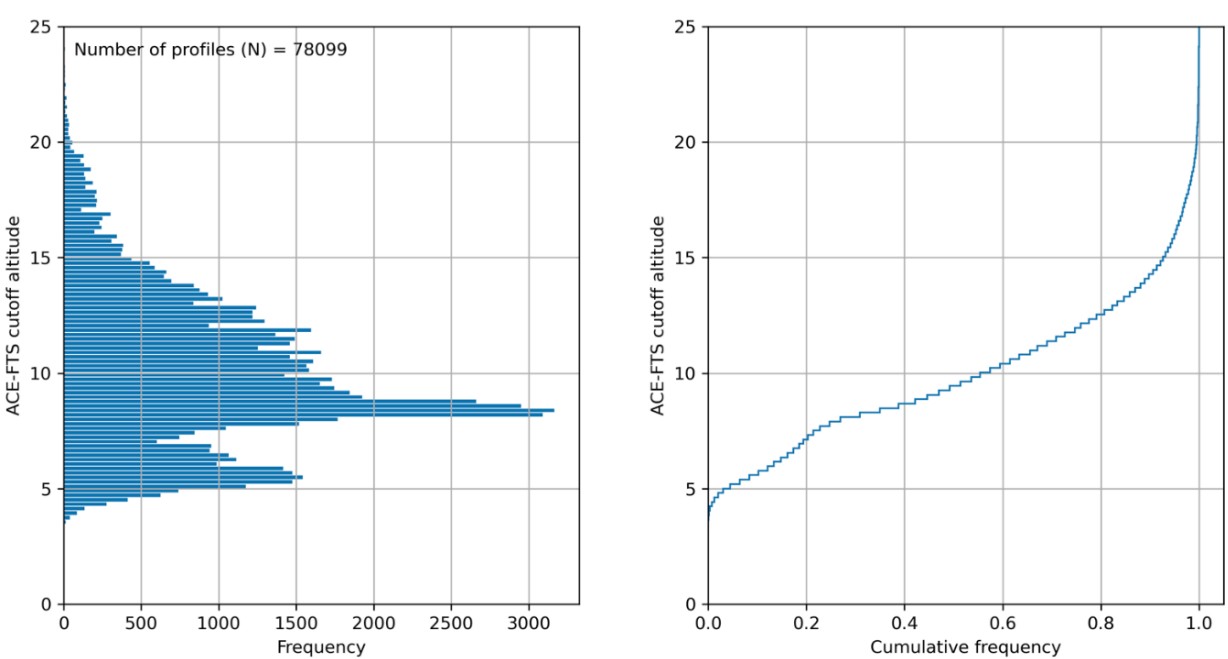

**Figure 10: Frequency distribution of the number of MAESTRO profiles with the corresponding ACE-FTS cutoff altitude (left panel) and corresponding cumulative frequency distribution (right panel).**

We tested the potential impact of Rayleigh scattering correction by removing all the MAESTRO profiles that have FTS cutoff
altitudes higher than 10 km. Cumulative frequency shows that this removes nearly 45% of the MAESTRO profiles. We re-
gridded this trimmed data set and repeated the analysis. The new comparison metrics are plotted in Fig. 8. Correlations between
the trimmed MAESRO data set and SAGE III improve considerably between 17 and 20 km for most wavelengths. For example,
in the case of comparing MAESTRO to SAGE III extinction at 18 km and 1012 nm case shown in Fig. 7, trimming the
MAESTRO data based on the ACE-FTS cutoff altitude increases the correlation coefficient increases from 0.45 to 0.66. In
general, we find that applying the altitude cutoff threshold decreases the standard deviations of the gridded MAESTRO data.





This analysis leads us to the conclusion that accounting for the Rayleigh scattering contribution can lead to reduced variability in MAESTRO data.

## 5 Discussion and Conclusions

Observations from MAESTRO offer a potentially important dataset that fills the data gap in continuous multi-wavelength solar
occultation measurements of stratospheric aerosol extinction coefficients between the end of the SAGE II mission in 2005 and the start of the SAGE III mission in 2017. In this study, the quality of MAESTRO aerosol extinction measurements was investigated through comparison with measurements from SAGE II, SAGE III and OSIRIS. We find that, despite significant scatter in MAESTRO extinction and SAOD, gridded median MAESTRO aerosol extinction is in good agreement with SAGE III during background periods. After volcanic eruptions and wildfire injections of stratospheric aerosol, MAESTRO aerosol
extinction enhancements are well correlated with SAGE III, but biased low. This bias depends on the wavelength, and it decreases with increasing wavelengths in general. We note that the bias reported here in MAESTRO measurements is specific to the version 3.13 dataset and would likely be different with updated processing in the forthcoming data versions. An improved comparison during periods of enhanced aerosol extinction is obtained by tuning MAESTRO extinctions with SAGE III data during their overlap period using a power-law fit at different altitudes and wavelengths. We also checked to see if larger
variability in MAESTRO extinction measurements could be explained by variation in observing latitude, or aspects of the observation geometry. Although not shown here, no significant correlation with any of these variables was found in the lower stratosphere, which is the region most impacted by volcanic or wildfire events occurring in mid to high latitudes, where the sampling from the MAESTRO is most frequent. It was found that some of this variability is related to the Rayleigh scattering correction scheme, and that it offers a promising avenue to improve the MAESTRO aerosol data in future data releases.

Information about stratospheric aerosol particle sizes in the lower stratosphere can be obtained from the MAESTRO AE values, but with lesser confidence due to reduced signal to noise ratio and low spectral correlation. The difference in spectral response of MAESTRO measurements, especially following major events may be a limiting factor in accurately characterizing stratospheric aerosol particle size information with MAESTRO. We find a long-term trend in the AE derived from multi-spectral MAESTRO aerosol extinction measurements in the NH, linked to changes in extinction measurements at shorter
wavelengths over the MAESTRO measurement period. Since this result is not consistent with AE derived from SAGE II and SAGE III, we suggest it is possibly related to MAESTRO instrumental artefacts. While this finding limits the use of MAESTRO AE to study long-term changes in aerosol size distribution, the MAESTRO data is useful in the exploration of short-term impacts of individual eruptions on particle size. For example, MAESTRO AE results suggest a decrease in aerosol size after the Raikoke eruption, which is consistent with many other eruptions but inconsistent with interpretation of data from
SAGE III for this particular eruption (Thomason et al. 2021, Wrana et al. 2023), which may be due to the different spatial



sampling of the two instruments, with MAESTRO potentially sampling the stronger aerosol perturbations poleward of the eruption.

This study shows that information from MAESTRO may be useful to complement other satellite records after carefully accounting for its uncertainties, especially at higher latitudes and during the data gap in SAGE records. It should be noted that the release of a new version (v4) of the MAESTRO aerosol data is forthcoming. Future work with this new version of the data is of priority to understand the impact of this update.

**Data Availability**

All data used in this study is freely available. MAESTRO and ACE-FTS data is available after registration from https://database.scisat.ca/level2/. OSIRIS data can be accessed from the University of Saskatchewan server at https://research-groups.usask.ca/osiris/data-products.php. SAGE II and SAGE III/ISS data is available after registration from the NASA Atmospheric Science Data Center at https://asdc.larc.nasa.gov/project/SAGE%20II and https://asdc.larc.nasa.gov/project/SAGE%20III-ISS respectively.

**Competing Interests**

The contact author has declared that none of the authors has any competing interests.

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
