# Peer review of "Assessment of ACE-MAESTRO v3.13 multi-wavelength stratospheric aerosol extinction measurements"

_EGUsphere, 2024_

## Community Comment (CC1)

**Overall**

The authors present a comparison of SAGE, OSIRIS, and MAESTRO extinction products (including derived products like stratospheric aerosol optical depth and Ångström exponent) and present a method of correcting MAESTRO extinction to better match SAGE III/ISS. They developed a model as a function of wavelength, altitude, and latitude to transform the MAESTRO data to better agree with SAGE III/ISS.

First, the good! I find this paper to be scientifically interesting, and I see potential utility in this method. If this product can be validated (or at least evaluated to determine the quality and consistency of its performance) then it could be a valuable tool in filling the "SAGE gap". I thank the authors for presenting this interesting study.

Second, the concerns. The paper would benefit from a thorough proofreading for the sake of clarity. It's *not* poorly written, there are just several sentences that do not make sense and several details that are not clear enough to enable the reader to reproduce the methodology. While I have tried to address some of these factors below, I suspect the authors will make several more corrections after a thorough proofreading.

I have 3 major concerns with this paper as written:

1. The authors used v5.2 of the SAGE III/ISS data product while v5.3 is available. This is not a big deal, per se. However, v6.0 is scheduled for release in January 2025 and the changes between v5.3 and v6.0 are aerosol-centric. For example, the so-called seagull effect (i.e., a low bias in the 520, 601, and 676 nm channels) has been nearly completely eradicated. This will undoubtedly impact some of the authors' corrections.

2. The authors tended to focus more on lower altitudes (e.g., 12-18 km, Figs. 8 & 9) where the seagull effect was less prominent. If the authors insist on using v5.2 or v5.3 then it would greatly benefit the reader to see how your correction method performed at altitudes that are more susceptible to the seagull (i.e., 20 - 30 km).

3. The paper would benefit from a more systematic and thorough presentation of how this method improves the agreement between MAESTRO and the other instruments (including more discussion with the agreement with SAGE II). I think this is a missed opportunity to highlight the impact of your work. Something that shows how the overall statistics (e.g., median percent difference, median absolute deviation, etc.) improved (even breaking the samples into "background" and "elevated" aerosol conditions). I don't want to dictate how this is done, these are just suggestions; I am happy to discuss possibilities offline if you prefer.

**General Conclusion**

This is an interesting paper and I believe it should be published after the authors have had an opportunity to make some modifications. While I listed several concerns with the paper above, and more detailed concerns/corrections below, I do not believe any of these would preclude publication. If I could insist on 1 change to this paper it would be the improvement suggested in the 3rd bullet above. Why? Because this will provide the greatest benefit to the reader for understanding the value of the authors' work. That said, I accept that these changes will be made at the authors' and editor's discretion.

**Specific Comments**

Line 52 you state that particle size distribution information is available within multi-wavelength occultation measurements. This is true, but this is generally overly simplified in the literature and too often important details are neglected. We addressed many of these challenges, and the impact of various assumptions, within a recently-published paper that you may find interesting and useful (https://amt.copernicus.org/articles/17/2025/2024/).

Line 136: Why not use v5.3? It probably makes minimal difference for the aerosol product, but if you have a reason for 5.2 over 5.3 it would be interesting to hear. As an aside, v6.0 will soon be released (January 2025), which has noticeable changes to the aerosol product (e.g., the "seagull" effect has been largely corrected) and you may wish to re-evaluate your correction factors after that release.

Did you make any corrections to the SAGE III/ISS 520 nm channel? Ray Wang showed an offset in that channel as well as 601 and 676 nm (Fig. 3 in https://doi.org/10.1029/2020JD032430) and we discussed our correction method using a simple power-law correction (Eq. 2 in https://doi.org/10.5194/amt-15-5235-2022). I am not saying these papers must be cited; I provide them for your reference.

Line 159: The 603 and 675 nm channels (520 to a lesser extent) for SAGE III are significantly impacted by ozone and should be used cautiously. Maybe this does not impact your analysis, but it's worth noting. This should all be resolved in v6.0.

Starting on line 159: "During background stratospheric conditions..." I do not understand what the author's are trying to communicate here. Please consider rewording?

Line 173: "...525 nm is excluded, details in Sect. 3.1..." It is unclear why the 525 nm channel is excluded and Sect. 3.1 does not shed light on this either (at least it is not clear to me). This channel looks no worse than any other; why exclude it? Can you please clarify in the paper?

Fig. 3: First off, the authors highlighted the differences at low altitudes (e.g., around 12 km, or so). I am not so concerned about this; did you filter for clouds? If not, then I suspect cloud contamination. Mahesh Kovilakam has a cloud-filter algorithm for SAGE III/ISS (https://amt.copernicus.org/articles/16/2709/2023/) that should be part of the v6.0 release in January (it is currently available in a separate repository cited in his paper) and could be used in your current analysis.

Fig. 3, continued: My main concern with this figure is the similar behavior between the 525/603/779 channels. The SAGE 602 channel was significantly impacted by ozone in v5.2 and v5.3 (biased low by 10-40% depending on latitude/altitude), so I would expect the percent difference in your plot to favor MAESTRO (i.e., MAESTRO should be greater than SAGE...or at least less negative), but it looks like the percent difference becomes *more* negative. Further, the offset changes sign for 1021 nm throughout much of the atmosphere. Overall, this makes it difficult to interpret and I wonder how well behaved the MAESTRO extinction spectra are (sorry, but I know almost nothing about MAESTRO). Can the authors please comment on this? Should we expect the aerosol spectrum to be better behaved?

Fig. 3, final comment: The color scales are difficult to read (this applies to all contour/meshgrid plots in the paper). The authors claimed that the coefficient of correlation was 0.6, but I cannot infer this from the figures. Have you evaluated different color maps and/or limits? As is, I cannot interpret these plots because I cannot distinguish between the various shades. I realize creating these types of figures is quite challenging, so I am sorry to complain about this, but I have a very hard time reading these. Just a suggestion.

Line 257ff: "As aerosol content of the stratosphere varies..." This sentence reads as if composition plays no role. We demonstrated the impact of composition (smoke and sulfuric acid content) in our paper (https://amt.copernicus.org/articles/17/2025/2024/) and Chris Boone demonstrated variability in the sulfuric acid content (https://doi.org/10.1016/j.jqsrt.2023.108815). Because your analysis involves background, volcanically perturbed, and major wildfire events, this sensitivity should be acknowledged.

Fig. 6 It is unclear how the AE was calculated. I understand this was discussed in Eq. 1 in Section 2.3, but I did not find that section to be helpful in understanding what you did. Could the authors please clarify how the AE was calculated (was it multi-spectral?) and what wavelengths were used?

Line 396: "We note that the bias reported here in MAESTRO measurements is specific to the version 3.13 dataset and would likely be different with updated processing in the forthcoming data versions." This raises several questions:

- When is the next version scheduled for release?

- Do you have any estimate on how much this may change?

- Why not delay publication until you have the next MAESTRO version *and* v6.0 of SAGE III/ISS?

Line 405: "Information about stratospheric aerosol particle sizes in the lower stratosphere can be obtained from the MAESTRO AE values..." This will be challenging because of MAESTRO's limited spectral range. Our group inferred PSD values from SAGE II using 2 channels (525/1020; https://amt.copernicus.org/articles/17/2025/2024/) and it does not appear unreasonable, but the certainty definitely goes down. You may be able to use MAESTRO's 755 nm channel to tease out some more information, but this would require a lot of caution. I am just urging caution with your current statement. However, if you only intend to make general statements (e.g., the particles got "bigger" or "smaller" after an eruption), then what you have is a generally safe statement. Would you please clarify?

---

## Author Comment (AC1)

Responses to Community comments

Thank you to Dr. Knepp for these comments. Our responses are listed in blue below.

The authors present a comparison of SAGE, OSIRIS, and MAESTRO extinction products (including derived products like stratospheric aerosol optical depth and Angstrom exponent) and present a method of correcting MAESTRO extinction to better match SAGE III/ISS. They developed a model as a function of wavelength, altitude, and latitude to transform the MAESTRO data to better agree with SAGE III/ISS.

First, the good! I find this paper to be scientifically interesting, and I see potential utility in this method. If this product can be validated (or at least evaluated to determine the quality and consistency of its performance) then it could be a valuable tool in filling the "SAGE gap". I thank the authors for presenting this interesting study.

Thank you!

Second, the concerns. The paper would benefit from a thorough proofreading for the sake of clarity. It's not poorly written, there are just several sentences that do not make sense and several details that are not clear enough to enable the reader to reproduce the methodology. While I have tried to address some of these factors below, I suspect the authors will make several more corrections after a thorough proofreading.

We do believe our edits in response to the reviewer and community comments have improved the readability of the manuscript, and we have proofread the revised manuscript to improve word choice in many instances.

I have 3 major concerns with this paper as written:

1. The authors used v5.2 of the SAGE III/ISS data product while v5.3 is available. This is not a big deal, per se. However, v6.0 is scheduled for release in January 2025 and the changes between v5.3 and v6.0 are aerosol-centric. For example, the so-called seagull effect (i.e., a low bias in the 520, 601, and 676 nm channels) has been nearly completely eradicated. This will undoubtedly impact some of the authors' corrections.

We have performed a quick check to see how using SAGE III v5.3 data would change our analysis. The figure below shows the parameters of a power law fit between MAESTRO and the two versions of the SAGE III data (v5.2 and v5.3). It shows that there is a nearly identical result with either v5.3 or v5.2. We checked other wavelengths and they also show negligible difference between the two SAGE III data versions.

It is interesting to learn that v6.0 data is now available. It will be interesting to include version 6 in future work. That being said, we predict that changes in SAGE III/ISS aerosol retrievals will be relatively small compared to the large biases we have highlighted in the MAESTRO data (up to a factor of two difference compared to SAGE III under perturbed conditions), and we suggest that since the aim of our study is to assess the broad utility of the MAESTRO data, and that changing the version of SAGE III/ISS in our comparison is unlikely to change the main conclusions of our study.

[Figure]

2. The authors tended to focus more on lower altitudes (e.g., 12-18 km, Figs. 8 & 9) where the seagull effect was less prominent. If the authors insist on using v5.2 or v5.3 then it would greatly benefit the reader to see how your correction method performed at altitudes that are more susceptible to the seagull (i.e., 20 - 30 km).

Indeed, as discussed in the manuscript, we did focus on lower altitudes where aerosol extinction has its largest signal, and where MAESTRO inter-wavelength correlations show greatest values. We understand some readers may be interested in the comparisons at higher altitudes, and have updated Fig. 9 (revised manuscript) with the vertical axis extended to 30 km compared to 25 km before.

3. The paper would benefit from a more systematic and thorough presentation of how this method improves the agreement between MAESTRO and the other instruments (including more discussion with the agreement with SAGE II). I think this is a missed opportunity to highlight the impact of your work. Something that shows how the overall statistics (e.g., median percent difference, median absolute deviation, etc.) improved (even breaking the samples into "background" and "elevated" aerosol conditions). I don't want to dictate how this is done, these are just suggestions; I am happy to discuss possibilities offline if you prefer.

We agree this is useful idea, and we have addressed this comment with two additional figures.

First, Fig. 2 of the revised manuscript shows the results of a more traditional comparison between MAESTRO and SAGE III, based on collocated measurements, for the full overlap period, as well as for a subset representative of "background" aerosol conditions. This plot and the discussion related to it provides evidence of the scatter in the MAESTRO raw measurements, and motivates the use of robust metrics like the median.

In terms of quantifying the impact of the correction procedure, we include Fig. 10 in the updated manuscript, also reproduced below. This figure shows the result of using a "coincident" data comparison of MAESTRO with SAGE III, and shows the median percent difference before and after application of the correction scheme. The correction works best for the 603 and 675 nm wavelengths, reducing the percent bias down to less than about 25% below 24 km. While at longer wavelengths the correction works less well, it generally decreases the median bias at all altitudes and generally reduces the median bias to less than approximately 30%. Further work may well improve upon our methods, but we believe this is a useful proof of concept for the MAESTRO data correction we have introduced here.

[Figure]

Figure 10: Coincident comparison of MAESTRO aerosol extinction measurements with SAGE III. Lines show the median of differences computed as (MAESTRO – SAGE III)/SAGE III. Black lines show the median of differences for the raw data, while green lines show the same for the data after the correction of Sec. 4.1 is applied to the MAESTRO data

General Conclusion

This is an interesting paper and I believe it should be published after the authors have had an opportunity to make some modifications. While I listed several concerns with the paper above, and more detailed concerns/corrections below, I do not believe any of these would preclude publication. If I could insist on 1 change to this paper it would be the improvement suggested in the 3rd bullet above. Why? Because this will provide the greatest benefit to the reader for understanding the value of the authors' work. That said, I accept that these changes will be made at the authors' and editor's discretion.

Specific Comments

Line 52 you state that particle size distribution information is available within multi-wavelength occultation measurements. This is true, but this is generally overly simplified in the literature and too often important details are neglected. We addressed many of these challenges, and the impact of various assumptions, within a recently-published paper that you may find interesting and useful (https://amt.copernicus.org/articles/17/2025/2024/).

Thank you for pointing us to this study which does very nicely spell out the challenges involved in retrieving particle size information from spectral measurements. We have modified the text here to acknowledge the challenges involved with reference to the suggested paper.

Line 136: Why not use v5.3? It probably makes minimal difference for the aerosol product, but if you have a reason for 5.2 over 5.3 it would be interesting to hear. As an aside, v6.0 will soon be released (January 2025), which has noticeable changes to the aerosol product (e.g., the "seagull" effect has been largely corrected) and you may wish to re-evaluate your correction factors after that release.

As written above, we have checked and see that the change in version 5.2 to 5.3 does not affect our comparison. While the changes included in version 6 may quantitatively affect the comparisons, we are confident that such changes will not change our main qualitative conclusions, which are that while there are substantial challenges with the MAESTRO data, that there is useful information therein. We hope that future work will be able extend our work by including new data versions.

Did you make any corrections to the SAGE III/ISS 520 nm channel? Ray Wang showed an offset in that channel as well as 601 and 676 nm (Fig. 3 in https://doi.org/10.1029/2020JD032430) and we discussed our correction method using a

simple power-law correction (Eq. 2 in https://doi.org/10.5194/amt-15-5235-2022). I am not saying these papers must be cited; I provide them for your reference.

We have not applied any correction to the SAGE III data. We have added text in Sec. 2.1.4 to make this clear in the manuscript.

Line 159: The 603 and 675 nm channels (520 to a lesser extent) for SAGE III are significantly impacted by ozone and should be used cautiously. Maybe this does not impact your analysis, but it's worth noting. This should all be resolved in v6.0.

These biases in SAGE III are now included in Sec. 2.1.4 and discussed again in the conclusions.

Starting on line 159: "During background stratospheric conditions…" I do not understand what the author's are trying to communicate here. Please consider rewording?

We have reorganized these sentences to more clearly communicate the issue here. The text now reads "The small differences in wavelength values between the two instruments is not expected to produce significant differences in the extinction values: during background stratospheric conditions (relatively undisturbed by volcanic eruptions or wildfires), the difference in extinction is expected to be less than 6% for the pair having the largest separation in wavelengths (779 nm and 756 nm) and less than 3% for all other wavelength pairs."

Line 173: "…525 nm is excluded, details in Sect. 3.1…" It is unclear why the 525 nm channel is excluded and Sect. 3.1 does not shed light on this either (at least it is not clear to me). This channel looks no worse than any other; why exclude it? Can you please clarify in the paper?

The MAESTRO bias compared to SAGE III varies non-monotonically with height compared to the other wavelengths—also the correlation between the two instruments is rather weaker at 525 nm compared to the other wavelengths. We have edited the text here as well as in Sec. 3.1 to make these points and the motivation for excluding 525 nm from the AE calculation more clear.

Fig. 3: First off, the authors highlighted the differences at low altitudes (e.g., around 12 km, or so). I am not so concerned about this; did you filter for clouds? If not, then I suspect cloud contamination. Mahesh Kovilakam has a cloud-filter algorithm for SAGE III/ISS (https://amt.copernicus.org/articles/16/2709/2023/) that should be part of the v6.0 release in January (it is currently available in a separate repository cited in his paper) and could be used in your current analysis.

Cloud screening is not part of the MAESTRO data processing chain, nor is screening explicitly applied to the MAESTRO aerosol retrievals used in this work. This point is included in our description of the MAESTRO aerosol data (Sec. 2.1.1), and whether or not cloud screening is used for comparison data sets is listed in the respective descriptions. Nonetheless, since many of our comparisons rely on median values within latitude and month bins, it is likely that many of the retrievals strongly affected by clouds are excluded from our analysis. This point is mentioned in Sec 2.3.

If MAESTRO data were to be used in future scientific studies, application of a cloud screening procedure, for example similar to that described by Kovilakam et al. (2024) should be investigated to better understand their impact.

Fig. 3, continued: My main concern with this figure is the similar behavior between the 525/603/779 channels. The SAGE 602 channel was significantly impacted by ozone in v5.2 and v5.3 (biased low by 10-40% depending on latitude/altitude), so I would expect the percent difference in your plot to favor MAESTRO (i.e., MAESTRO should be greater than SAGE…or at least less negative), but it looks like the percent difference becomes more negative. Further, the offset changes sign for 1021 nm throughout much of the atmosphere. Overall, this makes it difficult to interpret and I wonder how well behaved the MAESTRO extinction spectra are (sorry, but I know almost nothing about MAESTRO). Can the authors please comment on this? Should we expect the aerosol spectrum to be better behaved?

This work is really the first to do any systematic comparison of MAESTRO aerosol extinction with other data sets. Due to some challenges with the instrument and data analysis— which we now mention explicitly in the manuscript—we are not terribly surprised by the fact there is significant scatter in the MAESTRO data and biases with respect to other instruments. We hope that this work will motivate future work to look more carefully at the sources of bias and improve the MAESTRO retrieval or post-processing.

Fig. 3, final comment: The color scales are difficult to read (this applies to all contour/meshgrid plots in the paper). The authors claimed that the coefficient of correlation was 0.6, but I cannot infer this from the figures. Have you evaluated different color maps and/or limits? As is, I cannot interpret these plots because I cannot distinguish between the various shades. I realize creating these types of figures is quite challenging, so I am sorry to complain about this, but I have a very hard time reading these. Just a suggestion.

The colorbars in Figure 4 (revised manuscript) have been adjusted to make it easier to read the magnitudes of the anomalies plotted.

Line 257ff: "As aerosol content of the stratosphere varies…" This sentence reads as if composition plays no role. We demonstrated the impact of composition (smoke and sulfuric acid content) in our paper (https://amt.copernicus.org/articles/17/2025/2024/) and Chris Boone demonstrated variability in the sulfuric acid content (https://doi.org/10.1016/j.jqsrt.2023.108815). Because your analysis involves background, volcanically perturbed, and major wildfire events, this sensitivity should be acknowledged.

We have edited the beginning of this section to acknowledge that indeed aerosol composition (especially wildfire smoke content) will affect the AE in addition to size distribution. This section mostly discusses the AE itself, without interpretation of it as a reflection of the size distribution. Only in the final two sentences of the section do we link our AE calculations with prior studies who have interpreted spectral changes with size distributions.

Fig. 6 It is unclear how the AE was calculated. I understand this was discussed in Eq. 1 in Section 2.3, but I did not find that section to be helpful in understanding what you did. Could the authors please clarify how the AE was calculated (was it multi-spectral?) and what wavelengths were used?

We have edited the text after Eq. 1 to clarify our method:

Extinction measurements at five wavelengths (e.g., 603, 675, 779, 875 and 1012 nm for MAESTRO) are used to calculate AE for MAESTRO and SAGE III respectively at each altitude of each profile, by performing an ordinary least squares regression of $\ln(\beta)$ on $\ln(\lambda)$: the slope of this regression is the Ångström exponent.

Line 396: "We note that the bias reported here in MAESTRO measurements is specific to the version 3.13 dataset and would likely be different with updated processing in the forthcoming data versions." This raises several questions:

• When is the next version scheduled for release?

• Do you have any estimate on how much this may change?

• Why not delay publication until you have the next MAESTRO version and v6.0 of SAGE III/ISS?

MAESTRO version 4 is now available. However, while ozone and a total optical depth product are available, aerosol extinction is not. A data description document mentions this is because of noted biases in the extinction data. Therefore, the version 3.13 aerosol extinction data described in our manuscript is the most up-to-date publicly available MAESTRO aerosol product. We do not know when or if there will be a subsequent aerosol

product released. We hope that our study will encourage the investment of further resources into MAESTRO-related work.

Line 405: "Information about stratospheric aerosol particle sizes in the lower stratosphere can be obtained from the MAESTRO AE values..." This will be challenging because of MAESTRO's limited spectral range. Our group inferred PSD values from SAGE II using 2 channels (525/1020; https://amt.copernicus.org/articles/17/2025/2024/) and it does not appear unreasonable, but the certainty definitely goes down. You may be able to use MAESTRO's 755 nm channel to tease out some more information, but this would require a lot of caution. I am just urging caution with your current statement. However, if you only intend to make general statements (e.g., the particles got "bigger" or "smaller" after an eruption), then what you have is a generally safe statement. Would you please clarify?

We have modified the start of this final paragraph to be more clear: we mean only that the AE calculated from MAESTRO shows some physically plausible variations after eruptions and wildfires, indicating there may be some value to the multispectral MAESTRO extinction data. Actual retrieval of size distribution parameters and the associated uncertainties in those parameters is certainly still a potential task for future work.

References:

Knepp, T. N., Thomason, L., Kovilakam, M., Tackett, J., Kar, J., Damadeo, R., and Flittner, D.: Identification of smoke and sulfuric acid aerosol in SAGE III/ISS extinction spectra, Atmos. Meas. Tech., 15, 5235–5260, https://doi.org/10.5194/amt-15-5235-2022, 2022.

Kovilakam, M., Thomason, L., and Knepp, T.: SAGE III/ISS aerosol/cloud categorization and its impact on GloSSAC, Atmos. Meas. Tech., 16, 2709–2731, https://doi.org/10.5194/amt-16-2709-2023, 2023.

Wang, H. J. R., Damadeo, R., Flittner, D., Kramarova, N., Taha, G., Davis, S., Thompson, A. M., Strahan, S., Wang, Y., Froidevaux, L., Degenstein, D., Bourassa, A., Steinbrecht, W., Walker, K. A., Querel, R., Leblanc, T., Godin-Beekmann, S., Hurst, D., and Hall, E.: Validation of SAGE III/ISS Solar Occultation Ozone Products With Correlative Satellite and Ground-Based Measurements, J. Geophys. Res. Atmos., 125, e2020JD032430, https://doi.org/10.1029/2020JD032430;PAGE:STRING:ARTICLE/CHAPTER, 2020.

---

## Author Comment (AC2)

Dear Editor, reviewers,

We thank the reviewers for their valuable comments on our submitted manuscript. Based on their comments, we have made a number of changes to the manuscript which we believe improve the clarity of the presentation and provide more evidence to support the conclusions. Below we include replies to each reviewer comment; reviewer comments are in black, and our replies in blue.

**Reviewer Comment 1**

The manuscript presents stratospheric aerosol observation from the MAESTRO, with comparison to SAGE and OSIRIS. Such comparison for MAESTRO have, to my knowledge, not been published before. The instrument provides data at high latitudes - regions where SAGE III-ISS does not cover. The manuscript is well-written and it is easy to follow the presentation and discussions. Comparing stratospheric aerosol observations among different instruments is important for providing more reliable data to the research community and to compile stratospheric aerosol climatologies for simulations. I would like to see more discussion about reasons behind the discrepancy between the datasets.

Regarding the comparison of AOD and aerosol extinction coefficients, it is not unexpected to see large differences in data from different satellite instruments. Do you have any estimate on the impact of patchiness in aerosol concentrations on the MAESTRO observations? To retrieve aerosol data one must assume homogeneous conditions throughout the entire line-of-sight, as far as I understand it.

The instrument's line of sight (semi-horizontal) requires a transformation algorithm that computes the conditions at the tangent point. In patchy conditions, the tangent altitude of the retrieved aerosol ext coef. will be lower than the real altitude of aerosol layers. Do you have an estimate on how large this displacement may be? If so, how does this relate to the same phenomenon for the SAGE instruments? It should lead to spread in the data of your comparisons.

Solar occultation instruments cannot quantify aerosol extinction coefficients when the AOD in the line-of-sight is high, leading to a low bias in the data. Is this limit for MAESTRO the same as SAGE's, and if not, does that have an impact on discrepancies in the AOD comparisons of the two instruments? Do they have similar challenges with missing data, leading to similar underestimates of the AOD when sampling optically dense aerosol? Is

this why there is a factor of 2 difference between the instruments after wildfire or volcanic events, and why it has a low bias in general?:

- L196 "…MAESTRO underestimates peak extinction values after major volcanic eruptions and wildfires by a factor of 2 or more…"
- L213: "…MAESTRO extinction at shorter wavelengths has a low bias of 40-80% compared to SAGE III nearly everywhere in the lower stratosphere except right above the tropical tropopause region…"
- L248 "…corresponds to a relative underestimation of 32% by MAESTRO…"
- L406: "…The difference in spectral response of MAESTRO measurements, especially following major events may be a limiting factor in accurately characterizing stratospheric aerosol particle size information with MAESTRO….".

This is important to know for a user, it is important for the reader to understand during which conditions the data may be useful, and it is important to know this when compiling an aerosol climatology like CREST or GloSSAC.

We thank the reviewer for these valuable comments. Concerning the source of the bias compared to other instruments, we hope that this study can provide evidence that can be used to identify the source(s) of discrepancies, but unfortunately, at present we can only speculate. Since MAESTRO and SAGE-III/ISS both utilize the solar occultation method, we do not believe that "saturation" effects due to the strong extinction would preferentially affect MAESTRO measurements, especially for the perturbations due to eruptions and wildfires seen during the overlap period. (A loss of signal due to strong extinction was an issue for SAGE III in the period following the 1991 Pinatubo eruption, but that was at least an order of magnitude larger aerosol perturbation compared to anything since.) To our knowledge, the impact of aerosol patchiness on retrievals and inter-instrument comparisons is not well understood. A recent study has addressed the issue somewhat by comparing aerosol results from the limb-sounding instrument OMPS using two retrieval techniques, a typical retrieval and one that utilizes a tomographic technique to better resolve two-dimensional structures the aerosol field, and suggest that non-uniformity can potentially lead to biases in the retrieved aerosol extinctions (Bourassa et al., 2023). However, in this case, since the biases appear to persist beyond a few weeks (after which the aerosol cloud is typically assumed to be approximately zonally symmetric), we do not believe that aerosol inhomogeneity would be a major factor. A related issue is that differences in sampling within monthly latitude bins can lead to biases when comparing binned data sets (Toohey et al., 2013), however, we found that MAESTRO anomalies were generally not correlated with observing latitude (within the latitude bins), or aspects of the

observation geometry, suggesting such sampling biases are not a significant component of the overall biases. Similarly, since biases are not limited to the lower stratosphere, but extend throughout the vertical extent, we believe that while cloud issues may be an important consideration for the lowest altitudes, that there are other, more general issues leading to the overall systematic biases seen. Processing MAESTRO measurements is challenging, in part due to the presence of contaminants in the instrument optical path—a fact that we have now mentioned in the main text—which could be a reason for systematic biases in the MAESTRO aerosol retrievals. Pinpointing the origin of these biases is beyond the scope of this work. However, the aim of this paper was to assess the potential information content of the MAESTRO aerosol data, in essence to gauge whether further investment of resources into MAESTRO measurement processing might be warranted. We believe the work presented supports future investigation into these issues.

To address the reviewer's comment, we have included many of the above-mentioned points in the paper's conclusion and discussion section.

How was bias from ice-clouds treated? Data within 2-3 km of the tropopause may be affected by ice clouds to a large degree, suggesting the need to screen for signals from ice-crystals within at least 2-3 km above the tropopause + some kilometer extra (depending on instruments vertical resolution).

Cloud screening is not part of the MAESTRO data processing chain, nor is screening explicitly applied to the MAESTRO aerosol retrievals used in this work. This point is included in our description of the MAESTRO aerosol data (Sec. 2.1.1), and whether or not cloud screening is used for comparison data sets is listed in the respective descriptions. Nonetheless, since many of our comparisons rely on median values within latitude and month bins, it is likely that many of the retrievals strongly affected by clouds are excluded from our analysis. This point is mentioned in Sec 2.3.

If MAESTRO data were to be used in future scientific studies, application of a cloud screening procedure, for example similar to that described by Kovilakam et al. (2024) should be investigated to better understand their impact.

Figure 7. Which latitudes and times are shown here? Is it latitude binned monthly mean data? Part of the spread in data may come from periods with patchy conditions. Have you tried excluding the first 2-4 months after volcanic eruptions or wildfires when fitting the data to see if these effects may cause large spread in your' comparison?

Each data point represents the median value of extinction coefficients (MAESTRO on the y-axis and SAGE III on the x-axis) in a ten-degree latitude bin in a single month. The figure caption and description in the main text have been expanded to describe this more fully. We do not see a strong impact of excluding the first 2-4 months after eruptions or wildfires. This figure (Fig. 8 in revised manuscript) illustrates a point that we see generally within the data, which is that the spread in the comparison between the instruments is dominated by cases where SAGE III values are small or moderate and MAESTRO values are elevated. In contrast, when both SAGE III and MAESTRO values are elevated (after eruptions or wildfires), we see fairly good correlation between the instruments, but a systematic offset. This observation motivates our application of the "tuning" technique.

L302: I understand why you do not fit linearly, but why use a power law? Is it the best possible fit to use in this case?

Linear fits to the MAESTRO vs. SAGE III comparison plots were found to often not perform well. The nature of the extinction data is that its distribution can be strongly skewed, with a large number of small values and a smaller population of values orders of magnitude larger corresponding to the elevated conditions after eruptions or wildfires. Linear ordinary least square fits to such skewed data will tend to prioritize the fit to the larger values at the expense of the fits to the smaller values. As a result, a correction method built on linear fits to the data produced reasonable agreement for perturbed conditions but poor agreement for background conditions. In contrast, a power-law fit, which is equivalent to a linear fit to the logarithmically-scaled data, performed much better over the range of conditions present in the data set.

The following plot illustrates the impact of the linear vs. power-law fitting procedure in bias-correcting the MAESTRO data compared to SAGE III based on collocated measurements, comparable to Fig. 10 in the revised manuscript. This figure illustrates that the power-law fit more successfully corrected the biases of the MAESTRO aerosol extinction data.

[Figure]

Figure R1: Coincident comparison of MAESTRO aerosol extinction measurements with SAGE III. Lines show the median of differences computed as (MAESTRO – SAGE III). Black lines show the median of differences for the raw data, while green lines show the same for the data after the correction of Sec. 4.1 is applied to the MAESTRO data, and blue lines show the same when a correction based on a linear fit to the data is applied.

**Reviewer Comment 2**

This manuscript is well described to the application study of ACE-MAESTRO for stratospheric aerosol distribution. Although the analysis for stratospheric aerosol is not so complicated, this research topic is rare. For this reason, the manuscript is valuable for this scientific community.

However, some background and methodology parts are unclear. For detail:

1) In lines 81-84: This manuscript aims to evaluate the climatological stratospheric aerosol studies using MAESTRO. The MAESTRO data is weel known dataset and observation record is long. But, it is not clear that the previous studies did not use the MAESTRO data for

climate record. Could you explain the reason? In addition, do you have some additional working for MAESTRO data to use the climate record?

It is true that MAESTRO aerosol data has not been used much in prior studies, apart from a handful of studies looking at specific eruptions, a point we make in the introduction. This is likely partially due to the noisiness of the MAESTRO data product, which leads to poor comparisons with other instruments on a profile-to-profile basis. However, as our manuscript shows, when robust statistics like the median are used to remove the impact of outliers, the MAESTRO data shows reasonable correlation with SAGE III, indicating there is useful information in the data set.

2) Section 2.1.1: For the MAESTRO algorithm, some unclear points are existed. For example, the manuscript is not clearly explained why the MAESTRO data uses the pressure and temperature profiles as the algorithm input. In addition, for the P and T profiles, the ACE-FTS v3.5/3.6 is the reference data for MAESTRO in lines 103-104. Is the vertical resolution of P and T profiles enough to get the lapse rate tropopause height?

ACE-FTS pressure and temperature data are used in the MAESTRO retrieval to estimate the contribution of Rayleigh scattering to the measured extinction. This point has been added to the description of the MAESTRO retrieval method.

Validating ACE-FTS against aircraft measurements in the UTLS of the NH midlatitudes has shown that the vertical resolution of ACE-FTS data varies from about 3 km to less than 1 km (Hegglin et al., 2008). The tropopause height should be estimated with reasonable accuracy from such data, although we acknowledge that tropopause height uncertainty will contribute to uncertainty in the vertically integrated stratospheric aerosol optical depth (SAOD).

In addition, the multi-wavelength of aerosol extinction coefficient was retrieved by the MAESTRO. I wonder how this wavelength is determined.

The MAESTRO instrument is a spectrometer, measuring radiation across a broad spectral range during solar occultations. Aerosol extinction is retrieved at particular wavelengths where the contribution to the extinction from trace gases is relatively small. A sentence making this point is included in Sec. 2.1.1.

3) Section 2.1: In this study, several satellites are used, and these satellites' specifications are quiet different. So, I suggest that the author make table for summarizing the specification of satellites.

We appreciate the suggestion. However, the instrument descriptions in Sections 2.1.2-2.1.4 are quite short and use a fairly parallel structure, so that a reader searching for information should be able to identify it quickly. We believe a table would only repeat information and take up more space, and thus prefer to not include a new table.

4) Section 2.2: All the satellites in this study are solar-occulation observation. Therefore, the horizontal coverage is too wide to define the specific location. To define the reference location of the satellite datasets, how to be define? (center of the light optical path? or the lowest point?)

The horizontal location of a retrieval from a limb-viewing (occultation or limb-scattering) instrument is the tangent point—the point of closest approach of the line of sight to the Earth's surface. Since this point will have the largest air density in the line of sight due to the exponentially decreasing density in a spherical atmosphere, the extinction along the line of sight will be dominated by the constituents at this location. The horizontal resolution of limb sounding instruments is estimated to be around 500 km (McElroy et al., 2007). This information has been added to the MAESTRO description in Sec. 2.1.1.

In lines 148, could you explain the details for the significant gaps over the extra-tropics around winter months?

OSIRIS measures scattered sunlight, producing denser sampling compared to solar occultation instruments but is restricted to measuring within the sunlit portion of the atmosphere. This leads to a smaller number of measurements in the winter hemisphere. The text has been updated to explain the OSIRIS sampling in more detail.

5) Section 2.3: Many of the detailed data explanation and variable definitions are included in this section. So, I suggest that this section will be moved before the section 2.2.

We suggest that Sec 2.3, on the analysis methods used, works well here as it leads into the results section, and Sec 2.2 on instrument sampling, works well coming directly after the section introducing the instruments. We do not find any variable definitions included in Sec 2.3 that are needed to understand Sec 2.2, Therefore we prefer to retain the present order of the sections.

6) Section 3.2: This section is too simple. The manuscript is only explained the statistical number and qualitative figure explanation. Could you explain more detailed explanation inlcuding the regional characteristics of the stratopheric aerosol linked to the wildfire and volcanic events?

The characterization of these events and their radiative forcing, aerosol spread etc. have been well documented in prior studies (e.g, Kloss et al. 2020, Kloss et al., 2021, Khaykin et al., 2020). We have added sentences to the introduction to point interested readers to relevant prior studies. In Sec 3.2, our focus in the comparison is not on characterizing the aerosol events, but rather on the differences in the data sets. This we believe is consistent with the aim of the study and the scope of the journal.

7)Figure 6: I suggest that the manuscript has to include the detailed explanation of this figure, such as the reason of difference between two products.

The conclusions and discussion section includes a full paragraph on the Angstrom exponent results, which we have updated, pointing to the documented instrument contamination issues as a likely reason for the apparent trend in the MAESTRO AE timeseries.

8) Section 4.1 (Line 302): Please explain the detailed reason to use the 'y = ax^b' function.

Please see response to Rev 1 line 302 above.

**Reviewer Comment 3**

The manuscript assesses the aerosol extinction retrieval (v3.13) in the stratosphere and related products by the MAESTRO visible spectrometer onboard SCISAT. This sounds very valuable and interesting as the community needs more independent datasets for monitoring of aerosol properties, providing more comprehensive instrument comparisons and for building long-term records. The manuscript is well-written and clear. I think this study deserves publication in AMT after the following points are clarified.

I would suggest the authors indicate in which conditions (background stratospheric content, high aerosol loadings, seasons, etc.) and for which type of aerosol climatology MAESTRO dataset can be useful for the community.

We thank the reviewer for this encouragement to be more specific in terms of the implications of our work. We have added sentences to the conclusions more explicitly addressing how MAESTRO data could be useful in future studies, and specifically highlight that the tuned extinction values show good agreement with OSIRIS and SAGE III for all seasons and for both background and perturbed conditions. We also provide a recommendation on which wavelengths of MAESTRO data might be most useful.

In section 2.1 describing the individual datasets, I would suggest to give more quantitative information about the biases when available in the literature comparing various spaceborne observations of stratospheric aerosol extinction in different latitudinal bands and/or aerosol loadings (e.g. OSIRIS vs SAGEII, SAGEIII vs OMPS). This will allow the reader to better grasp the relevance of MAESTRO data through comparisons with more typically used records. This may be summarized in a table.

Thank you for this suggestion. We have added statements regarding validation of other instruments including estimations of biases when available, including for OSIRIS (Rieger et al., 2019) and SAGE III (Kalnajs and Deshler, 2022). We agree this helps a reader put the MAESTRO biases discussed here in context to other work.

Also, what about including OMPS as there is a product from University of Saskatchewan of high interest for the scientific community?

We have chosen to focus on comparing MAESTRO with SAGE III, since both are solar occultation instruments, and given the relative simplicity of this observation method compared to limb scattering observations like OMPS, SAGE instruments are treated as the "gold standard" against which to calibrate other instruments. The period of overlap between MAESTRO and SAGE III is shorter than that for OMPS, however there are multiple aerosol events during the SAGE III overlap that we are confident we are sampling a sufficient range of extinctions to produce a useful comparison. Also, since there are multiple versions of OMPS retrieval, we are hesitant to include a single version, and hesitant to draw attention away from our main aims by including multiple versions of OMPS.

To minimize the impact of the outliers and of high altitude clouds (especially in the tropics), what about considering SAOD from tropopause altitude + 1 km?

This is of course a possibility, but prior studies have pointed to the importance of aerosol below 15 km above the tropopause as a significant amount of the total SAOD (e.g., Ridley et al, 2014, Andersson et al., 2016), and if we are to assess the utility of MAESTRO data for SAOD merged product, we suggest it is important to assess the total SAOD. We find that the total SAOD is in relatively good agreement with SAGE III, providing a justification for future work.

The choice of monthly-binning is relevant to feed stratospheric aerosol databases for climate modelling purposes. I guess you tried to bin over shorter temporal grids (e.g.

weekly) to examine whether the short-term aerosol variability is consistent in MAESTRO data. Why and do you have an idea about the spread of SAOD and AE values at the weekly or 2-weekly scale?

As the reviewer points out, our main focus is on the applicability of MAESTRO data to monthly-mean, zonal mean databases for climate modelling or other uses, and so we have not extensively explored the data on smaller temporal scales. The MAESTRO data does exhibit considerable variability, a significant proportion of which is unlikely to be geophysical in origin but rather due to instrumental and retrieval limitations. For this reason, we have utilized robust statistics including the median value of the binned data. Such statistics require a suitable sample size in order to be reliable, and so moving to higher temporal resolution is likely to reduce the sample size within bins and therefore lead to a larger relative scatter in the MAESTRO medians, especially over the tropics and mid-latitudes.

Figure 2: some gaps can be observed in MAESTRO time series during volcanic periods especially around 2010. What is the explanation? Is it due to some saturation effect that could depend on the amplitude of each event?

There does appear to be some periods of decreased MAESTRO aerosol extinction observation frequency, particularly in 2009 and 2011. For example, while in most years there are over 100 observations over the globe in the month of June, in June 2009 there are zero retrieved profiles, and in June 2011 there are three. They do occur close in time to the eruptions of Sarychev (17 June, 2009) and Nabro (June 12, 2011), although we point out that the decreased retrieval frequency seems to begin before the eruptions in both cases, and that observations are less frequent around the globe, and not particularly in the NH for the case of the high latitude NH eruption of Sarychev. Also, we do not see a decrease in observation frequency associated with the 2015 Calbuco or 2019 Raikoke eruptions. For these reasons we do not believe the "gaps" are directly resulting from the eruptions and are more likely due to other issues in data collection and processing.

Do you have any idea about the impact of spatial sampling biases in the average of MAESTRO data? Does this vary with seasons and years? From Figure 1 it seems that the limited spatial coverage in tropical latitudes could largely affect the robustness of zonal means there.

It is possible that sampling affects the binned monthly statistics shown here. We have checked to see if MAESTRO differences with respect to SAGE III are correlated with

aspects of the sampling including the mean sampled latitude and aspects of the observation geometry, but have not found these things to be significant in explaining the offsets. Still, we suggest it is possible that sampling differences may lead to random error compared to other instruments, and have included a short discussion of this aspect in the conclusions, with reference to papers that have explored this issue for trace gas measurements (e.g., Toohey et al., 2013, Sofieva e a., 2014).

Figure 6: I suggest the evolution of AE between MAESTRO and SAGEIII to be plotted at 50°N and 50°S (e.g. AE vs time) where both datasets overlap. It is quite difficult to see the difference with the colors used in the figure. The signal seems to remain high and quite steady over the years following the eruption in MAESTRO whereas it shows more variability in SAGEIII (unless I am mistaken by the color scale). The 12-km altitude is very close to the tropopause and I am wondering if unscreened clouds can be part of the explanation. For SAGEIII do we have any idea about the enhancement in AE for 2020 where no particular volcanic/fire event occurred?

While line plots of AE at certain latitudes are useful to compare the SAGE III and MAESTRO results quantitively, here we also want to point out that the spatial coverage of the instruments is different and may contribute to the apparent differences in the timeseries. For example, while the SAGE III AE appears to decrease immediately after the Raikoke eruption, the MAESTRO AE increases over a latitude range that is notable further north than that of SAGE III. While the comparison here is qualitative, we feel that it is a useful first step to gauge the general characteristics of the MAESTRO AE product compared to SAGE III. A more quantitative comparison follows in Fig. 11 (of revised manuscript), where, as suggested we show a line plot of AE for MAESTRO and both SAGE II and SAGE III, and we see that the MAESTRO AE shows a long-term drift in value over the full mission of MAESTRO. This result calls into question the reliability of the MAESTRO AE over long periods, and dissuades us from spending too much effort quantifying differences to SAGE III here.

It is possible that clouds are affecting the results in the lowest altitudes, but our aim here is to assess the data as provided to see what potential information can be taken from it.

We speculate that the enhancement in SAGE III AE in 2020 is related to the 2019 Raikoke eruption. In Figure 5 (revised document), it appears that SAOD in the NH high latitudes is still elevated from Raikoke into the summer of 2020, suggesting that aerosol from the eruption may still be present and the size distribution affected.

Section 4.2: the correct quantification of Rayleigh scattering is apparently an important parameter in MAESTRO data variability. Could any offline calculation of Rayleigh scattering along MAESTRO line of sight using a raytracing model (including atmospheric refraction) rather than incorporating measurements bt ACE-FTS be helpful to reduce the effect?

For the Rayleigh calculations included in the MAESTRO retrieval algorithm, an essential ingredient is the profile of atmospheric density, since the extinction due to Rayleigh scattering is directly proportional to the density. Therefore, information about density (or equivalently, pressure and temperature) is needed for the retrieval algorithm, which includes a radiative transfer calculation along the line of sight. The present methodology uses pressure and temperature retrievals from the ACE-FTS instrument. This provides a measure of robustness due to the identical optical path of the two instruments through the atmosphere. Our goal is to assess the MAESTRO aerosol extinction results and their utility, and hope that our results may indeed lead to future improvements in the data processing.

L21, 278: Ulawun instead of Ulawan.

fixed

L346: MAESTRO instead of MAETRO.

 fixed

Some references are missing. Please check throughout the manuscript e.g. Kovilakam et al. (2023), Malinina et al. (2018), Randall et al. (2001), Sofieva et al. (2022, 2024), Thomason et aL. (2007).

For instance:

Mahesh Kovilakam, Larry Thomason, and Travis Knepp, SAGE III/ISS aerosol/cloud categorization and its impact on GloSSAC, Atmos. Meas. Tech., 16, 2709–2731, 2023, https://doi.org/10.5194/amt-16-2709-2023

Elizaveta Malinina, Alexei Rozanov, Vladimir Rozanov, Patricia Liebing, Heinrich Bovensmann, and John P. Burrows, Aerosol particle size distribution in the stratosphere retrieved from SCIAMACHY limb measurements, Atmos. Meas. Tech., 11, 2085–2100, https://doi.org/10.5194/amt-11-2085-2018, 2018

Fixed

In the reference list, the Vernier et al. (2011a) is not cited in the manuscript.

Fixed

Reply references:

Andersson, S. M., Martinsson, B. G., Vernier, J.-P., Friberg, J., Brenninkmeijer, C. A. M., Hermann, M., van Velthoven, P. F. J., and Zahn, A.: Significant radiative impact of volcanic aerosol in the lowermost stratosphere, Nat. Commun., 6, 7692, https://doi.org/10.1038/ncomms8692, 2015.

Bourassa, A. E., Zawada, D. J., Rieger, L. A., Warnock, T. W., Toohey, M., and Degenstein, D. A.: Tomographic Retrievals of Hunga Tonga-Hunga Ha'apai Volcanic Aerosol, Geophys. Res. Lett., 50, e2022GL101978, https://doi.org/10.1029/2022GL101978, 2023.

Hegglin, M. I., Boone, C. D., Manney, G. L., Shepherd, T. G., Walker, K. A., Bernath, P. F., Daffer, W. H., Hoor, P., and Schiller, C.: Validation of ACE-FTS satellite data in the upper troposphere/lower stratosphere (UTLS) using non-coincident measurements, Atmos. Chem. Phys., 8, 1483–1499, https://doi.org/10.5194/acp-8-1483-2008, 2008.

Kalnajs, L. E. and Deshler, T.: A New Instrument for Balloon-Borne In Situ Aerosol Size Distribution Measurements, the Continuation of a 50 Year Record of Stratospheric Aerosols Measurements, J. Geophys. Res. Atmos., 127, e2022JD037485, https://doi.org/10.1029/2022JD037485;PAGE:STRING:ARTICLE/CHAPTER, 2022.

Khaykin, S., Legras, B., Bucci, S., Sellitto, P., Isaksen, L., Tencé, F., Bekki, S., Bourassa, A., Rieger, L., Zawada, D., Jumelet, J., and Godin-Beekmann, S.: The 2019/20 Australian wildfires generated a persistent smoke-charged vortex rising up to 35 km altitude, Commun Earth Environ, 1, https://doi.org/10.1038/s43247-020-00022-5, 2020.

Kloss, C., Sellitto, P., Legras, B., Vernier, J. P., Jégou, F., Venkat Ratnam, M., Suneel Kumar, B., Lakshmi Madhavan, B., and Berthet, G.: Impact of the 2018 Ambae Eruption on the Global Stratospheric Aerosol Layer and Climate, J. Geophys. Res. Atmos., 125, e2020JD032410, https://doi.org/10.1029/2020JD032410, 2020.

Kloss, C., Berthet, G., Sellitto, P., Ploeger, F., Taha, G., Tidiga, M., Eremenko, M., Bossolasco, A., Jégou, F., Renard, J.-B., and Legras, B.: Stratospheric aerosol layer perturbation caused by the 2019 Raikoke and Ulawun eruptions and their radiative forcing, Atmos Chem Phys, 21, 535–560, https://doi.org/10.5194/acp-21-535-2021, 2021.

Kovilakam, M., Thomason, L., and Knepp, T.: SAGE III/ISS aerosol/cloud categorization and its impact on GloSSAC, Atmos. Meas. Tech., 16, 2709–2731, https://doi.org/10.5194/amt-16-2709-2023, 2023.

McElroy, C. T., Nowlan, C. R., Drummond, J. R., Bernath, P. F., Barton, D. V., Dufour, D. G., Midwinter, C., Hall, R. B., Ogyu, A., Ullberg, A., Wardle, D. I., Kar, J., Zou, J., Nichitiu, F., Boone, C. D., Walker, K. A., and Rowlands, N.: The ACE-MAESTRO instrument on SCISAT: Description, performance, and preliminary results, Appl Opt, 46, https://doi.org/10.1364/AO.46.004341, 2007.

Ridley, D. A., Solomon, S., Barnes, J. E., Burlakov, V. D., Deshler, T., Dolgii, S. I., Herber, A. B., Nagai, T., Neely, R. R., Nevzorov, A. V., Ritter, C., Sakai, T., Santer, B. D., Sato, M., Schmidt, A., Uchino, O., and Vernier, J. P.: Total volcanic stratospheric aerosol optical depths and implications for global climate change, Geophys. Res. Lett., 41, 7763–7769, https://doi.org/10.1002/2014GL061541;PAGE:STRING:ARTICLE/CHAPTER, 2014.

Rieger, L. A., Zawada, D. J., Bourassa, A. E., and Degenstein, D. A.: A Multiwavelength Retrieval Approach for Improved OSIRIS Aerosol Extinction Retrievals, Journal of Geophysical Research: Atmospheres, 124, https://doi.org/10.1029/2018JD029897, 2019.

Sofieva, V. F., Kalakoski, N., Päivärinta, S.-M., Tamminen, J., Laine, M., and Froidevaux, L.: On sampling uncertainty of satellite ozone profile measurements, Atmos. Meas. Tech., 7, 1891–1900, https://doi.org/10.5194/amt-7-1891-2014, 2014.

Toohey, M., Hegglin, M. I., Tegtmeier, S., Anderson, J., Añel, J. A., Bourassa, A., Brohede, S., Degenstein, D., Froidevaux, L., Fuller, R., Funke, B., Gille, J., Jones, A., Kasai, Y., Krüger, K., Kyrölä, E., Neu, J. L., Rozanov, A., Smith, L., Urban, J., von Clarmann, T., Walker, K. A., and Wang, R. H. J.: Characterizing sampling biases in the trace gas climatologies of the SPARC Data Initiative, J. Geophys. Res. Atmos., 118, 11,847-11,862, https://doi.org/10.1002/jgrd.50874, 2013.